# SWI/SNF chromatin remodeler complex within the reward pathway is required for behavioral adaptations to stress

Abdallah Zayed[1], Camille Baranowski[1], Anne-Claire Compagnion[1], Cécile Vernochet[1], Samah Karaki[1], Romain Durand-de Cuttoli [2], Estefani Saint-Jour[3], Soumee Bhattacharya[1], Fabio Marti[2,5], Peter Vanhoutte[3], Moshe Yaniv[4], Philippe Faure[2,5], Jacques Barik [1,6,7], Laurence Amar[1], François Tronche [1,8 ✉] & Sébastien Parnaudeau[1,8]

Enduring behavioral changes upon stress exposure involve changes in gene expression sustained by epigenetic modifications in brain circuits, including the mesocorticolimbic pathway. Brahma (BRM) and Brahma Related Gene 1 (BRG1) are ATPase subunits of the SWI/SNF complexes involved in chromatin remodeling, a process essential to enduring plastic changes in gene expression. Here, we show that in mice, social defeat induces changes in BRG1 nuclear distribution. The inactivation of the *Brg1/Smarca4* gene within dopamine-innervated regions or the constitutive inactivation of the *Brm/Smarca2* gene leads to resilience to repeated social defeat and decreases the behavioral responses to cocaine without impacting midbrain dopamine neurons activity. Within striatal medium spiny neurons, *Brg1* gene inactivation reduces the expression of stress- and cocaine-induced immediate early genes, increases levels of heterochromatin and at a global scale decreases chromatin accessibility. Altogether these data demonstrate the pivotal function of SWI/SNF complexes in behavioral and transcriptional adaptations to salient environmental challenges.

[1] Sorbonne Université, Gene Regulation and Adaptive Behaviors, Neuroscience Paris-Seine, IBPS. CNRS UMR8246, INSERM, Paris, France. [2] Sorbonne Université, Neurophysiology and Behaviors, Neuroscience Paris-Seine, IBPS. CNRS UMR8246, INSERM, UPMC, Paris, France. [3] Sorbonne Université, Neuronal Signaling and Gene Regulation, Neuroscience Paris-Seine, IBPS. CNRS UMR8246, INSERM, UPMC, Paris, France. [4] Developmental and Stem Cell Biology department, Pasteur Institute, Paris, France. [5] Present address: Brain Plasticity Unit, CNRS, ESPCI Paris, PSL Research University, Paris, France. [6] Present address: Université Côte d'Azur, Nice, France. [7] Present address: Physiopathology of Neuronal Circuits and Behavior, IPMC, Valbonne, France. [8] These authors contributed equally: François Tronche, Sébastien Parnaudeau. ✉email: francois.tronche@sorbonne-universite.fr

In the past decade, intensive efforts have been made to understand the mechanisms through which environmental challenges, such as stress exposure, can lead to life-long behavioral adaptations. These adaptations can be beneficial, for instance in adjusting social interactions to the behavior of congeners, but may also lead to the appearance of behavioral disorders. They are underpinned by long-lasting neuroadaptations which are thought to arise from changes in gene expression. In this context, the focus has naturally been brought onto epigenetic mechanisms and compelling evidence shows that transcription factors[1], but also histone modifications[2–4], and DNA methylations[5] within the meso-cortico-limbic system regulate the vulnerability to chronic stress exposure and the development of depressive-like behaviors in mice.

Stress exposure elicits a rapid rise of glucocorticoid hormones within the blood flow. Through their binding to the glucocorticoid receptor (GR), a ubiquitously expressed transcription factor, they play a key role in orchestrating the transcriptional response to stress exposure thereby facilitating one's ability to cope with environmental challenges. However, frequent solicitation of the stress response may lead to psychiatric diseases such as depression or addiction that are often associated with dysregulated glucocorticoids homeostasis[6]. Our previous work showed that pathological stress-responses triggered by exposure to chronic social stress involve GR in dopamine-innervated brain areas, which include the striatum, the nucleus accumbens (NAc), and deep-layers of the prefrontal cortex (PFC)[7]. We have also demonstrated that $GR/Nr3c1$ gene inactivation in those regions decreases behavioral responses to psychostimulants[8–10].

GR modulates gene expression through distinct mechanisms. It can either interact with components of the basal transcriptional machinery or recruit histone modifiers or chromatin remodeler complexes such as the mating-type switching/sucrose non-fermenting (SWI/SNF) complex[11,12]. These complexes use the energy of ATP to disrupt nucleosome DNA contacts, making possible the shift of nucleosomes along DNA and/or the removal or the exchange of nucleosomes. Consequently, they facilitate the accessibility of DNA or chromatin to proteins that can shape transcriptional responses[13]. Mammalian SWI/SNF are multisubunit complexes that contain either Brahma (BRM) or Brahma-related gene 1 (BRG1) as central ATPase subunits and 10–12 BRM/BRG1 associated factors[14].

While there is compelling evidence for the involvement of DNA- and histone-modifying enzymes in behavioral impairment induced by stress and drug exposure, very little is known regarding the potential role of chromatin remodeling complexes in such responses. The upregulation within the NAc of ATP-utilizing chromatin assembly and remodeling factor (ACF), a distinct chromatin remodeler, has been shown to be key for the development of stress-induced depressive-like behavior and for cocaine's reinforcing properties[15,16]. In addition, striatal BRG1 has been involved in cocaine relapse[17]. Here, we show that stress exposure induces a stable reorganization of GR and BRG1 nuclear distribution within striatal medium spiny neurons (MSNs). We further show that BRG1 in dopaminoceptive neurons, in neurons of the NAc, and BRM are essential for the development of social avoidance repeated social defeat in mice, a well-validated pre-clinical model of depression. Mice lacking either BRG1 in dopaminoceptive neurons, or BRM also exhibit decreased responses to cocaine in locomotor sensitization. Stress- and cocaine-induced behaviors are known to rely on the activity of dopamine neurons within the ventral tegmental area. The absence of BRG1 or BRM in our models does not alter dopaminergic activity, ruling out an effect through this cell population. Instead, it most probably relies on a cell autonomous defect, since the absence of BRG1 in striatal MSNs is associated with an impaired

stress-induced immediate early genes transcriptional response. This impairment is not due to an improper integration of glutamate and dopamine signals and the ensuing activation of the ERK signaling pathway within the MSNs but is associated with reduced Serine 10 phosphorylation of the histone 3. In addition, the absence of BRG1 in NAc MSNs led to an increase of transcriptionally silent heterochromatin levels and differential DNA accessibility at hundreds of loci distributed all over the genome.

Altogether, these results posit the SWI/SNF complexes within the mesocorticolimbic system as a key player in adaptive transcriptional responses to stress and drugs of abuse and the ensuing behavioral changes.

## Results

**GR and BRG1 nuclear distribution in NAc MSNs are affected by social defeat.** BRG1/BRM containing SWI/SNF complexes interact with GR in various tissues and cell lines[18–20]. We indeed observed an interaction between BRG1/BRM and GR in the brain by performing co-immunoprecipitation from striatal protein extracts (Fig. 1a). GR was detected in striatal lysates from control ($GR^{loxP/loxP}$) mice. Its level was reduced in striatal lysates prepared from $GR^{D1Cre}$ mice in which GR is depleted from most projecting medium spiny neurons (MSNs) (two first lanes respectively). We detected GR in protein extracts immunoprecipitated by a GR antibody (Fig. 1a, lane 3, control) as well as with anti-BRG1 (lane 4) or anti-BRM (lane 5) antibodies. To complement this biochemical approach, we next examined whether BRG1 and GR co-localize within MSN nuclei. Using co-immunostainings, we showed that BRG1 and GR proteins are present in cells nuclei of most brain regions including MSNs from the dorsal striatum and NAc (Fig. 1b). To study the topological distribution of GR and BRG1 within MSN nuclei and evaluate the proportion of GR potentially interacting with BRG1-containing complexes, we analyzed GR-BRG1 co-localization using confocal imaging. BRG1 and GR proteins distribute within nuclear foci. Under basal conditions, we observed an average of $80 \pm 9$ BRG1-containing foci (green) per MSN nucleus of the NAc, $247 \pm 24$ GR-containing foci (red) and a limited number of foci ($2.6 \pm 0.5$) containing both (yellow, Fig. 1c, basal condition and Fig. 1d). An acute social defeat, which elicits glucocorticoid release and GR activation, triggered a 1.7-fold increase in the number of GR foci per nucleus, one hour post defeat (Fig. 1c, acute social defeat and Fig. 1d left panel) and a 2.2-fold increase in the number of BRG1 foci (Fig. 1d middle panel). By contrast, twenty-four hours following ten days of daily social defeat, the numbers of foci for both GR and BRG1 were not significantly different from basal conditions although a trend toward an increase was observed for GR (Fig. 1c, repeated social defeat, and Fig. 1d left and middle panels respectively). The number of foci containing both proteins was higher after both an acute social defeat (3.5 folds) and repeated social defeats (3 folds, Fig. 1d). The latter result suggests a stable reorganization of GR and BRG1 nuclear distributions and points to a potential involvement of GR and BRG1 interaction in the long-term behavioral effects mediated by protracted stress exposure.

**$Brg1/Smarca4$ gene ablation in dopamine-innervated areas prevents behavioral adaptation induced by repeated social defeat.** Ten days of repeated social defeat induce enduring social aversion and anxiety-like (Fig. 2a, b). This paradigm is widely used as a pre-clinical model of depression[21]. We previously showed that GR inactivation in dopamine-innervated areas ($GR^{D1Cre}$ mice) prevents the appearance of social aversion after chronic social defeats[7]. We thus assessed the involvement of BRG1- or BRM-containing chromatin remodeling complexes in the appearance of long-term behavioral changes induced by stress

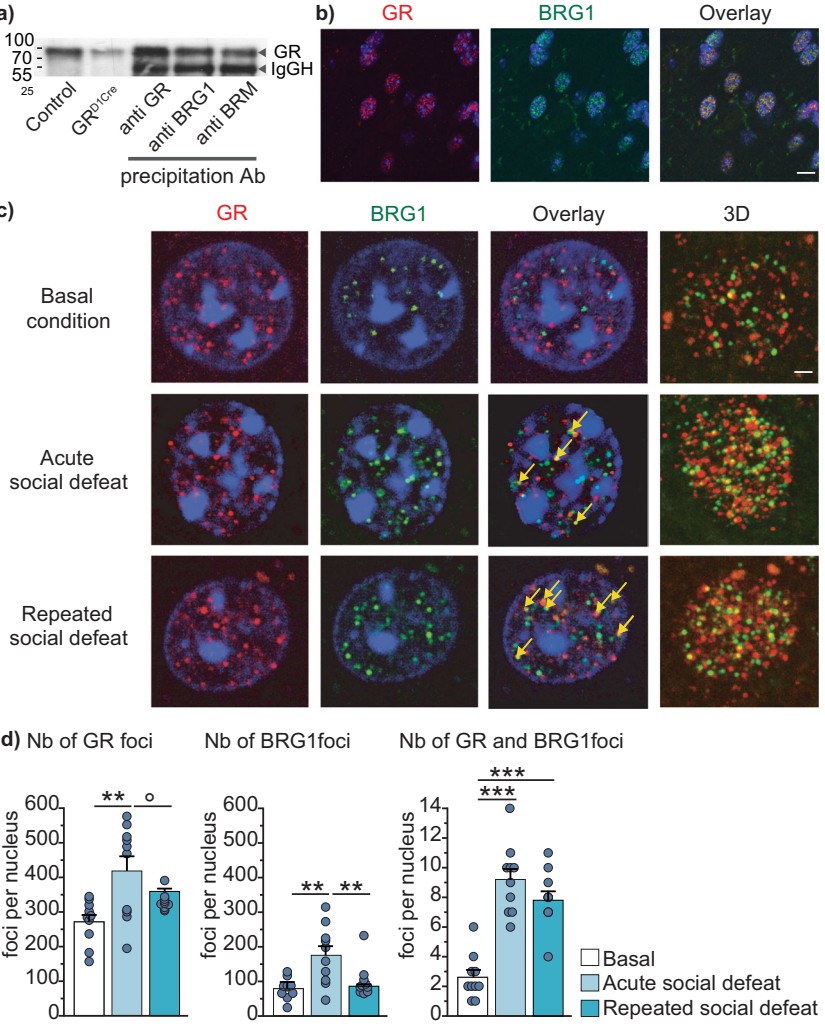

**Fig. 1 Colocalization of BRG1 and GR within the nucleus accumbens under basal conditions or after stress exposure. a** Co-immunoprecipitation of GR, BRG1, and BRM in whole striatum protein extracts. Lanes 1 and 2 correspond to GR detection in whole striatum lysates from $GR^{loxP/loxP}$ control mice and $GR^{D1Cre}$ mice respectively. Lanes 3, 4, and 5 correspond to GR detection after immunoprecipitation with GR, BRG1, and BRM antibodies, respectively. Left, molecular weight (kDa), IgGH, Immunoglobulin Heavy Chain **b** Co-staining of GR (in red), BRG1 (in green) and DNA (in blue, DAPI) in nuclei within the NAc. Scale bar = 20 µm. **c** High magnification of GR (red), BRG1 (green) and DAPI (blue) co-staining in a NAc MSN nucleus in an undefeated mouse (basal condition), 1 hour after an acute social defeat or 24 h after 10 days of repeated social defeats. The three first columns correspond to one focal for GR-DAPI, BRG1-DAPI and overlay stainings with the yellow bars pointing at foci in which BRG1 and GR colocalize. The last column corresponds to the nuclei 3D images. Scale bar = 2 µm. **d** Quantification of GR (left chart; treatment effect $F(2,29) = 8.91$, $p = 0.003$), BRG1 (middle chart; treatment effect $F(2,29) = 10.30$, $p = 0.0005$) and both GR and BRG1 colocalization foci (right chart; treatment effect $F(2,29) = 32.07$, $p < 0.0001$) in MSNs of the NAc from undefeated mice (basal), 1 h after an acute social defeat, or 24 hours after repeated (10 days) social defeat. $n = 10$ nuclei in 4 mice per condition. One-way ANOVA followed by Bonferroni correction, left panel: $°p = 0.0898$, $**p = 0.0024$; middle panel: $**$basal $vs$ acute $p = 0.0012$, $**$acute $vs$ repeated $p = 0.0023$; right panel $***p < 0.0001$. Data are represented as mean ± s.e.m. Data are provided in a Source Data file.

exposure. For this purpose, we generated mice deprived of BRG1 in the same cell populations as $GR^{D1Cre}$ mice (*i.e.*, dopamine-targeted neurons; $Brg1^{D1Cre}$ mice). Due to the potential overlap of BRG1 and BRM functions, we also used constitutive *Brm/Smarca2* mutant mice ($Brm^{-/-}$) to assess whether both factors act independently or can compensate each other. With that same aim in mind we also generated compound mutants bearing both mutations ($Brg1^{D1Cre}$:$Brm^{-/-}$). In control mice, both BRG1 and BRM proteins are present within MSN nuclei (Supplementary Fig. 1a, upper row). $Brg1^{D1Cre}$ mice showed a gene recombination profile coherent with that expected when using the YAC-D1aCre transgenic line[7,22], with a deletion of *Brg1* in around 90% of MSNs in the dorsal striatum and the NAc (Supplementary Fig. 1a, middle row), as well as in pyramidal neurons of deep cortical layers. In $Brm^{-/-}$ mice, BRM was totally absent from all cells

(Supplementary Fig. 1a, lower row). Of note, we observed reciprocal but partial compensation of *Brg1* and *Brm* expression in MSNs of our mutant models. Indeed, we found a 1.5-fold increase of BRM immunostaining mean intensity in $Brg1^{D1Cre}$ MSNs ($n = 10$; t-test p < 0.01), and a 1.3-fold increase of BRG1 in $Brm^{-/-}$ MSNs ($n = 12$–$13$; t-test $p < 0.001$, Supplementary Fig. 1b). Although BRG1-containing SWI/SNF complexes are known to control the proliferation rate and the differentiation process of various cell types[14], we observed no gross cytoarchitectural alteration and no difference in the number of neurons within the striatum and the cortex in any of our models, probably due to the fact that Brg1 deletion occurs in post-mitotic neurons (Supplementary Fig. 1a, Nissl staining and Supplementary Fig. 1c, d).

In absence of social defeat, $Brg1^{D1Cre}$ and their control littermates exhibited significant social interaction as they spent

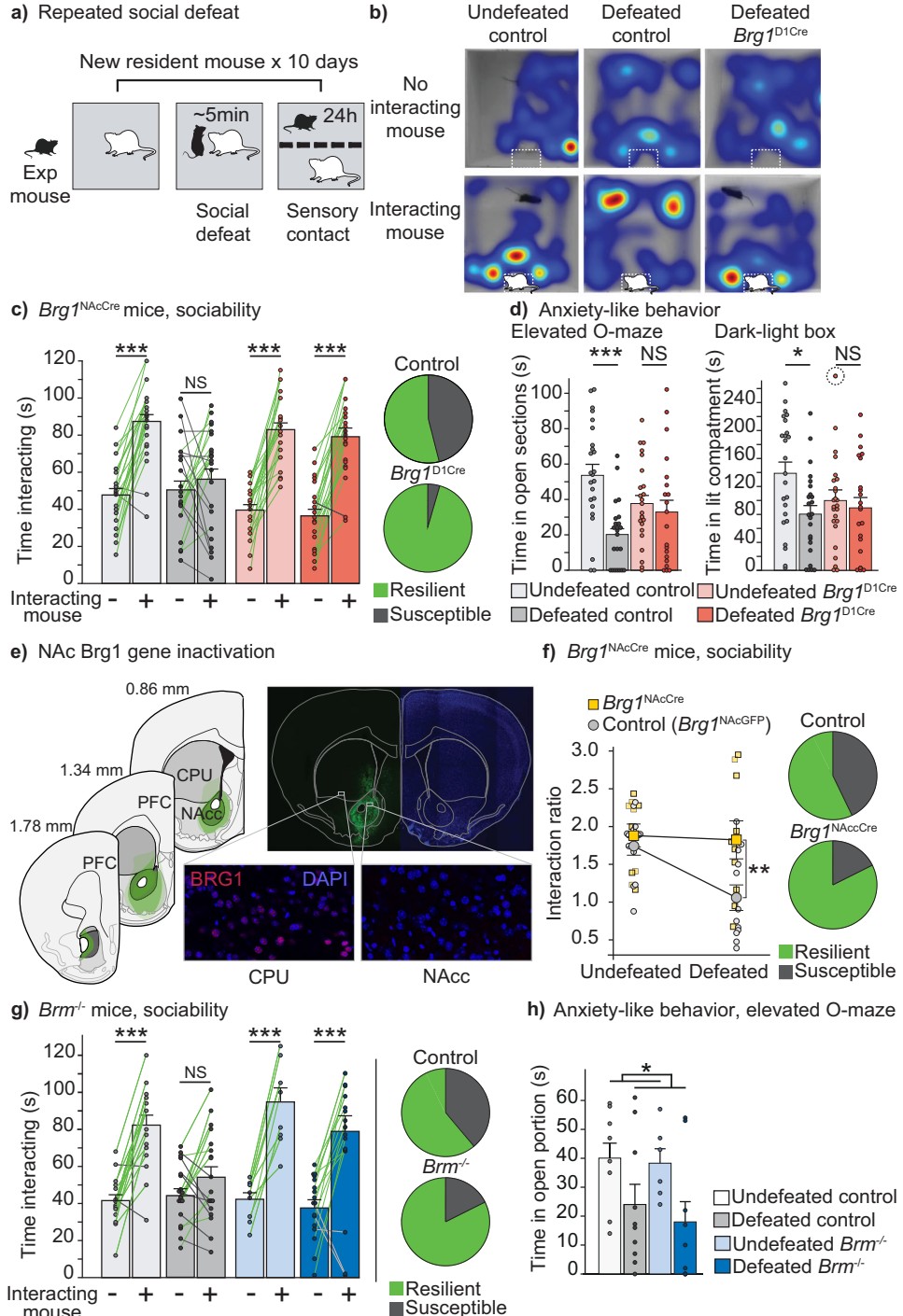

**a)** Repeated social defeat

**b)** Undefeated control / Defeated control / Defeated *Brg1*$^{D1Cre}$

**c)** *Brg1*$^{NAcCre}$ mice, sociability

**d)** Anxiety-like behavior
Elevated O-maze / Dark-light box

**e)** NAc Brg1 gene inactivation

**f)** *Brg1*$^{NAcCre}$ mice, sociability

**g)** *Brm*$^{-/-}$ mice, sociability

**h)** Anxiety-like behavior, elevated O-maze

more time interacting with an unfamiliar mouse than in the vicinity of an empty box. As expected, repeated social defeats abolished this social preference in control mice (Fig. 2b, c dark grey columns). Strikingly, mice deprived of BRG1 in dopamine-innervated areas kept interacting normally with the unfamiliar mouse despite a history of social defeats (Fig. 2b, c red columns). As previously observed[7], almost half of control mice (11 out of 24) exhibited social avoidance (interaction time with an empty box equal or higher than with a CD1 unfamiliar mouse). In stark contrast, in *Brg1*$^{D1Cre}$ mice only one animal out of 22 displayed social aversion after repeated defeats (Fig. 2c). In other words, the depletion of BRG1-containing SWI/SNF complexes in dopamine-innervated areas clearly favors a state of resilience. This difference

in sociability was confirmed in another display, the 3 chambers test, using C57Bl/6 mice as social targets (Supplementary Fig. 2a). Repeated social defeat increased anxiety-like behavior, as observed in defeated control mice when compared to undefeated controls (Fig. 2d). In the elevated O-maze test, *Brg1*$^{D1Cre}$ mice showed a trend toward more anxiety under basal conditions but did not exhibit that increase after repeated social defeat (Fig. 2d, left panel). The same result was found using the dark-light box test (Fig. 2d, right panel). There was no difference in locomotor activity between stressed and unstressed mice of both control and Brg1$^{D1Cre}$ groups (Supplementary Fig. 2b). Finally, social defeat is known to impact hormonal stress response. We thus measured the weight of adrenal glands as a proxy for average glucocorticoid

**Fig. 2 Behavioral effects of repeated social defeat in control, $Brg1^{D1Cre}$, $Brg1^{NAcCre}$ and $Brm^{-/-}$ mice. a** Repeated social defeat paradigm. **b** Representative hitmaps (red color means increased spent time) of social interaction in undefeated controls (left), defeated controls (middle) and defeated $Brg1^{D1Cre}$ (right). **c** Effect of repeated social defeat on social interaction time in control and $Brg1^{D1Cre}$ mice. Controls undefeated/defeated $n = 25/24$, $Brg1^{D1Cre}$ undefeated/defeated $n = 24/22$. Interaction social target x genotype x stress $F(1,91) = 10.25$; $p = 0.0019$. Three-way ANOVA with repeated measure followed by Bonferroni correction ***$p < 0.0001$. NS: $p > 0.99$. Pie charts: percentage of resilient (green) and susceptible (gray) animals in control (up) and $Brg1^{D1Cre}$ mice (bottom). $X^2(1,44) = 10.15$; $p = 0.0014$. **d** Effect of repeated social defeat on anxiety in controls and $Brg1^{D1Cre}$ mice. Control undefeated/defeated $n = 25/24$, $Brg1^{D1Cre}$ undefeated/defeated $n = 25/22$. Elevated-O-maze main stress effect $F(1,92) = 11.67$; $p = 0.0009$. Interaction genotype x stress $F(1,92) = 6.69$; $p = 0.0113$. Two-way ANOVA followed by Bonferroni correction ***$p < 0.0001$, NS: $p > 0.99$. Dark-Light box main stress effect $F(1,91) = 5.71$; $p = 0.019$. Two-way ANOVA followed by Bonferroni correction *$p = 0.0104$, NS: $p > 0.99$. A point out of scale (340 s) is pictured circled. **e** Transgene expression after viral injection along the rostro-caudal axis of the NAc (right panel). Dark green: minimal spreading; light green: maximal spreading; intermediate green: representative spreading pattern. Representative picture of GFP expression in the NAc (top right panel). Pictures showing the expression of BRG1 (in red) in the dorsal striatum of a $Brg1^{NAcCre}$ mouse (bottom left) and a lack of BRG1 expression in the NAc attesting for correct recombination of $Brg1$ gene (bottom right). **f** Effect of repeated social defeat on social interaction time in control ($Brg1^{NAcGFP}$, n = 12) and $Brg1^{NAcCre}$ mice ($n = 10$). Main stress effect $F(1,20) = 6.537$; $p = 0.0188$. Main group effect $F(1,20) = 4.534$; $p = 0.0459$. Interaction group x stress $F(1,20) = 4.904$; $p = 0.0386$. Interaction group x stress $F(1,20) = 4.904$;. Two-way ANOVA with repeated measure followed by Bonferroni correction **$p = 0.009$. Interaction group x stress $F(1,20) = 4.904$;. Two-way ANOVA with repeated measure followed by Bonferroni correction **. Pie charts: percentage of resilient (green) and susceptible (gray) anima ls in control (up) and $Brg1^{NAcCre}$ mice (bottom). $X^2(1,20) = 5.51$; $p = 0.0189$. **g** Effect of repeated social defeat on social interaction time in control and $Brm^{-/-}$ mice. Control undefeated/defeated $n = 17/18$, $Brm^{-/-}$ undefeated/defeated $n = 9/17$. Interaction social target x genotype $F(1,57) = 7.82$; p < 0.01. Interaction social target x stress $F(1,57) = 11.38$; $p = 0.0013$; three-way ANOVA with repeated measure followed by Bonferroni correction ***$p < 0.0001$. NS = not significant. Pie charts: percentage of resilient (green) and susceptible (gray) animals in control (up) and $Brg1^{D1Cre}$ mice (bottom). $X^2(1,33) = 1.93$. **h** Effect of repeated social defeat on anxiety in controls and $Brm^{-/-}$ mice. Control undefeated/defeated $n = 8/10$, $Brm^{-/-}$ undefeated/defeated n = 6/9. Main stress effect $F(1,29) = 7.37$; *$p = 0.0111$; two-way ANOVA. Data are represented as mean± s.e.m. Data are provided in a Source Data file.

production. As we previously observed[7], repeated social defeat induced a significant increase of adrenal gland weight; the increase was identical in both control and $Brg1^{D1Cre}$ mice (Supplementary Fig. 2c). In a similar way, general body weight increased in stressed individuals. No difference was observed between stressed $Brg1^{D1Cre}$ and control littermates (Supplementary Fig. 2d).

As mentioned, recombination of $Brg1$ in mutant mice encompasses distinct brain regions including the cortex, striatum and NAc. To narrow down the brain region in which BRG1 plays a role in stress-induced social impairment, and since the NAc has been repeatedly shown to be key for this, we proceeded to local gene inactivation in the NAc using virally mediated expression of the Cre recombinase. We performed bilateral injections of an AAV1-hSyn-Cre within the NAc of $Brg1^{L2/L2}$ adult mice ($Brg1^{NAcCre}$) while we injected a GFP-expressing virus to generate controls ($Brg1^{NAcGFP}$; see Methods). Post-behavioral histological analysis revealed an effective transgene expression and recombination of $Brg1$ in the NAc, with a minimal impact onto nearby regions such as the dorsal striatum (Fig. 2e). Before stress exposure, both controls and $Brg1^{NAcCre}$ mice exhibited similar normal social interaction with an interaction ratio (time spent with another mouse/time spent near the empty box) higher than 1 (Fig. 2f). While repeated defeats decreased sociability in controls with 7 out of 12 mice showing social avoidance, only 2 mice deprived of BRG1 in the NAc out of 10 exhibited social avoidance. In those behavioral paradigms $Brg1^{NAcCre}$ mice were phenocopies of $Brg1^{D1Cre}$ ones. This strongly suggests that among the dopaminoceptive neurons deprived of BRG1 in this later mutant, the dopaminoceptive neurons of the NAc are essential for susceptibility to repeated social defeat.

**Constitutive $Brm$ gene ablation prevents social avoidance induced by repeated social defeat.** Mice constitutively deprived of BRM also exhibited a normal social interaction despite the repeated social defeats like what was observed for the BRG1 conditional mutant models (Fig. 2g). Only 3 out of 17 $Brm^{-/-}$ mice showed social avoidance after repeated social defeat compared to 7 out of 18 for the controls. Concerning the anxiety-like following repeated social defeat, mice deprived of BRM displayed a different

phenotype than that of $Brg1^{D1Cre}$ mice. $Brm^{-/-}$ mice exhibited an increase of anxiety after repeated social defeats comparable to that of control mice whereas $Brg1^{D1Cre}$ individuals did not (Fig. 2h).

Unsurprisingly, $Brg1^{D1Cre}$:$Brm^{-/-}$ mice baring the two mutations also displayed significant social interaction after social defeat with only 1 mouse out of 7 showing social avoidance (compared to 6 out 9 for the controls) (Supplementary Fig. 3a). Under basal conditions, these compound mutants showed a bimodal distribution in the time spent in the lit compartment of a dark-light box test with some mice never exploring that area while others spent most of their time in it. Although we found no statistical difference in the time spent in the lit compartment, compound mutants took significantly more time to exit the dark portion of the box, which may indicate increased anxiety-like (Supplementary Fig. 3b). It is noteworthy that the depletion of both BRM and BRG1 led to a significant decreased locomotor activity which may prevent clear interpretation of some of the behavioral phenotypes observed in this model (Supplementary Fig. 3c).

Altogether, our data demonstrate that SWI/SNF remodeling complexes in dopaminoceptive neurons and in the NAc are key for the long-term behavioral adaptation induced by chronic stress-exposure. Previous studies have shown that BRG1-associated factors including BAF53b and BAF170 in mice[23–26] and BAP60 in drosophila[27,28] were key for neuronal plasticity, short-term and long-term spatial memory, and fear memory. We thus probed for potential changes in fear memory in our models which could potentially explain the resilient phenotype observed after repeated social defeats and found no difference in either contextual or cued fear memory (Supplementary Fig. 4a).

We also tested other types of learning and memory relying on the striatum and on the cortex that are heavily recombined in $Brg1^{D1Cre}$ mice. In a motor learning task, the rotarod, both $Brg1^{D1Cre}$ and $Brm^{-/-}$ mice showed normal and identical learning and performance compared to controls (Supplementary Fig. 4b). The same result was observed in a prefrontal-dependent T-maze non-matching to sample working memory task (Supplementary Fig. 4c). In contrast, compound mutants showed severe impairment in both motor learning and working memory probably due to the almost complete lack of functional SWI/

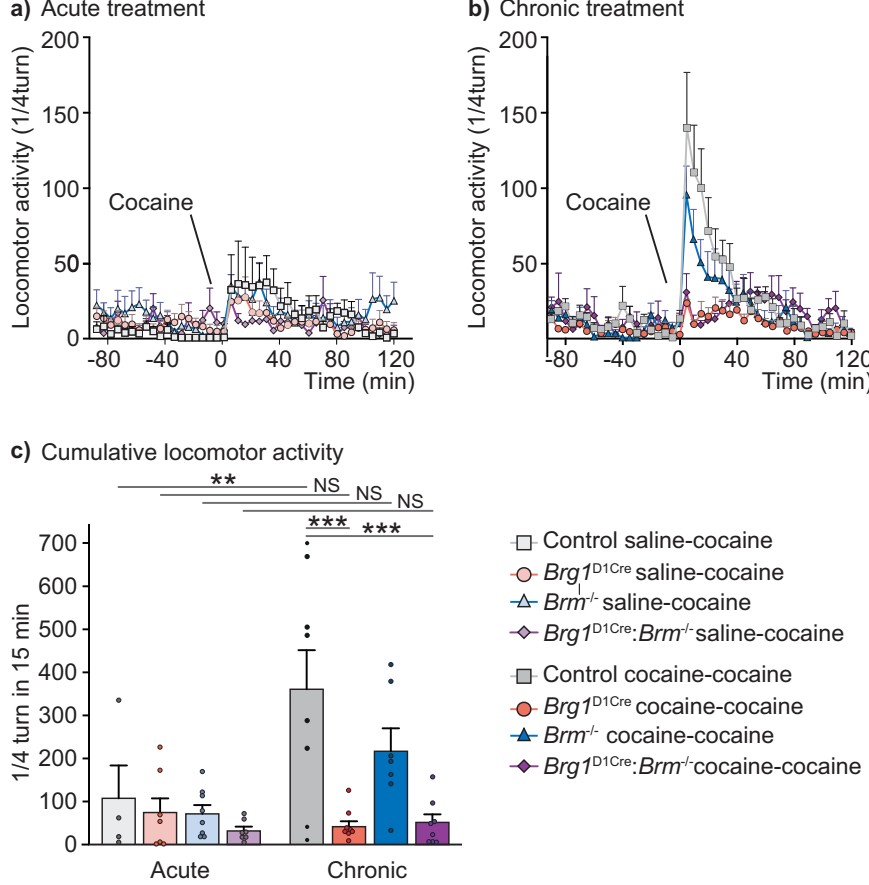

**Fig. 3 Behavioral sensitization to cocaine in control, *Brg1*^D1Cre, *Brm*^-/- and *Brg1*^D1Cre:*Brm*^-/- mice. a** Locomotor response (by 5 min blocks) to a cocaine injection (10 mg/kg) in saline pre-treated controls ($n = 4$), *Brg1*^D1Cre ($n = 7$), *Brm*^-/- ($n = 8$) and *Brg1*^D1Cre:*Brm*^-/- ($n = 7$) mice. **b** Locomotor response (by 5 min blocks) to a cocaine injection (10 mg/kg) in cocaine pre-treated control ($n = 8$), *Brg1*^D1Cre ($n = 8$), *Brm*^-/- ($n = 7$) and *Brg1*^D1Cre:*Brm*^-/- ($n = 8$) mice. **c** Locomotor sensitization (cumulated activity during 15 min after injection) to cocaine in control (acute $n = 4$; chronic $n = 8$), *Brg1*^D1Cre (acute $n = 7$; chronic $n = 8$), *Brm*^-/- (acute $n = 8$; chronic $n = 7$), and *Brg1*^D1Cre:*Brm*^-/- (acute $n = 7$; chronic $n = 8$) mice. Main drug effect $F(1,49) = 8.19$; $p = 0.0062$. Interaction genotype x drug $F(3,49) = 3.43$; $p = 0.0241$. Two-way ANOVA followed by Bonferroni correction acute *vs* chronic **$p = 0.0059$, NS = not significant ($p > 0.99$ for *Brg1*^D1Cre and *Brg1*^D1Cre:*Brm*^-/-, $p = 0.123$ for *Brm*^-/-); control *vs* *Brg1*^D1Cre ***$p < 0.0001$; control *vs* *Brg1*^D1Cre:*Brm*^-/- ***$p < 0.0001$. Data are represented as mean ± s.e.m. Data are provided in a Source Data file.

SNF complexes in key brain regions regulating these functions such as the dorsal striatum (Supplementary Fig. 4b, c).

**BRG1 in dopamine-targeted neurons modulates responses to cocaine.** In humans, stress-exposure is a well-known risk factor for drug addiction and many studies in animal models have shown that behavioral responses to drugs are sensitive to stress exposure and glucocorticoids[29,30]. Since SWI/SNF complexes are key for behavioral adaptation to social defeat, and that aversive and rewarding events share overlapping mechanisms of action, we next investigated their role in cocaine responses. We first examined the locomotor sensitization that occurs following repeated injections of cocaine. We showed that the locomotor response to an acute injection of 10 mg/kg of cocaine in saline pre-treated animals was similar between controls and *Brg1*^D1Cre mice (Fig. 3a, c). Daily injection of 10 mg/kg cocaine induced a significant locomotor sensitization in control mice as they showed an enhanced activity after a challenge injection of cocaine compared to saline pre-treated animals. However, the same treatment failed to induce any locomotor sensitization in *Brg1*^D1Cre mice, which displayed locomotor responses like the one elicited by an acute challenge (Fig. 3b, c). The complete absence of sensitization in *Brg1*^D1Cre mice was like that of the compound mutants suggesting that this behavioral response to cocaine strongly relies on

BRG1 containing complexes. The absence of BRM seemed to affect sensitization although to a lesser extent. Indeed, *Brm*^-/- mutant mice failed at showing a significant increase of locomotor response after chronic cocaine treatment though a trend could be observed (Fig. 3a–c).

To further investigate the involvement of SWI/SNF complexes into behavioral responses to cocaine, we also assessed cocaine-rewarding properties in a conditioned place preference paradigm. During pre-conditioning, mice showed no preference for any of the compartment of the place preference box (Supplementary Fig. 5, left panel) confirming an unbiased design. After conditioning, control mice developed a classical place-preference for the compartment associated with cocaine compared to the one associated with saline. Although *Brg1*^D1Cre mice also showed a place preference for the compartment associated with cocaine injections, it was significantly lower than the one of control mice thus showing an involvement of BRG1 reward processes (Supplementary Fig. 5, right panel). Overall, our results show that SWI/SNF complexes in dopamine-innervated areas are required for the behavioral changes induced by drugs of abuse. In addition, as observed for the social aversion following repeated social defeat, BRG1 and BRM appear to modulate cocaine sensitization although not with the same robustness. Conditioned place-preference in *Brm*^-/- mice was not assessed but may yield

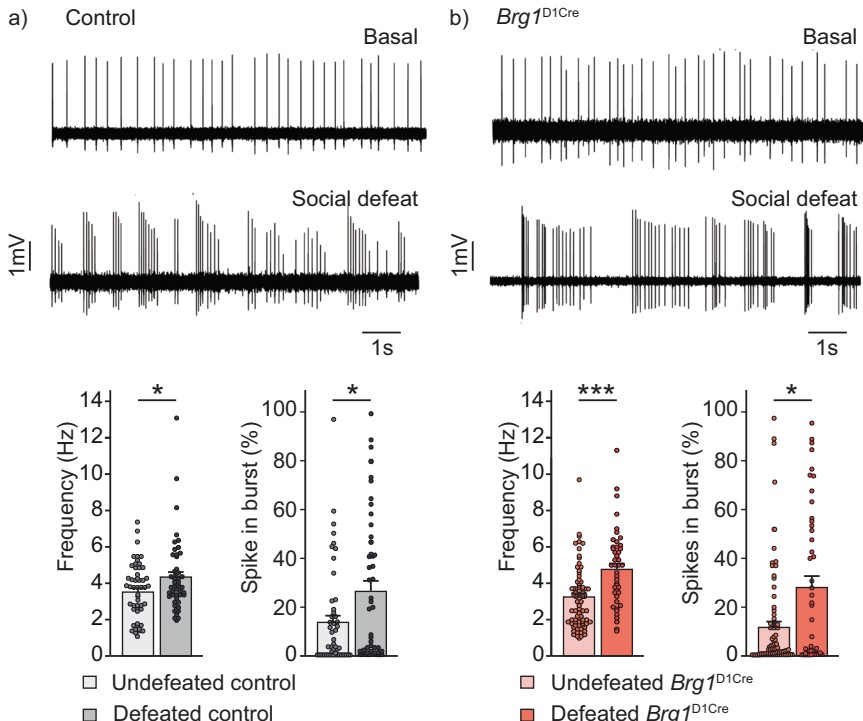

**Fig. 4 Consequences of repeated social defeat on ventral tegmental area dopamine neurons spontaneous activity in control and Brg1**[D1Cre] **mice.**
**a** Representative traces of dopamine neurons activity in undefeated (upper trace) and defeated (lower trace) control mice. Firing frequency (left chart) and percentage of spike in burst (right chart) in undefeated ($n = 51$ neurons) and defeated ($n = 51$ neurons) control mice. Two-sided T-test (for frequency) and Mann-Whitney test (spike in burst) *$p = 0.0244$ and $p = 0.0407$ respectively. **b** Representative traces of dopamine neurons activity in undefeated (upper trace) and defeated (lower trace) Brg1[D1Cre] mice. Firing frequency (left chart) and percentage of spike in burst (right chart) in undefeated ($n = 81$ neurons) and defeated ($n = 45$ neurons) Brg1[D1Cre] mice. Two-sided T-test (for frequency) and Mann-Whitney test (spike in burst) ***$p < 0.0001$, *$p = 0.0222$. Data are represented as mean ± s.e.m. Data are provided in a Source Data file.

similar result (*i.e.*, a lighter decrease than in Brg1[D1Cre] mice) as sensitizing and rewarding properties often co-vary.

**Dopaminergic firing in response to chronic social defeat in Brg1**[D1Cre] **mice.** Repeated social defeat induces a long-lasting increase of firing and bursting activities of dopamine neurons in the ventral tegmental area[31]. We and others previously showed that this increase is causally linked with the appearance of social aversion[7,32]. We thus examined whether the inactivation of BRG1 within dopaminoceptive neurons could also impact VTA cellular adaptations observed after social defeat. We thus measured dopamine neuron activity within the VTA of control and Brg1[D1Cre] mice in basal conditions or after chronic social defeat. In control mice, social defeat induced a significant increase of dopamine neurons firing rate (from $3.5 \pm 0.2$ Hz in unstressed animals to $4.3 \pm 0.3$ Hz in stressed animals) and a significant increase of bursting activity ($13.8 \pm 2.8\%$ of spike in burst in unstressed *vs* $26.5 \pm 4.2$ % in stressed animals) (Fig. 4a). Surprisingly, a similar basal activity and the same increase of dopamine neuron activity was observed in Brg1[D1Cre] mice with a firing rate going from $3.2 \pm 0.2$ Hz to $4.8 \pm 0.3$ Hz and a percentage of spike in burst going from $11.8 \pm 2.3\%$ to $28.1 \pm 4.7\%$ after social defeat (Fig. 4b and S6a). Basal firing of dopamine neurons was also unchanged in both Brg1[D1Cre] and Brm[-/-] mice compared to respective control animals (Supplementary Fig. 6a, b). However, compound mutant mice deprived of both BRG1 in dopaminoceptive neurons and BRM exhibited under basal conditions a marked reduction of dopamine neurons firing rate and bursting activity compared to controls (Supplementary Fig. 6c), an observation similar to that observed in GR mutant animals[8]. Compensatory mechanisms between BRM and BRG1 are

therefore probably at play for the regulation of dopamine cells firing. Altogether, these results indicate that the behavioral phenotypes observed in absence of BRG1 within dopaminoceptive neurons are unlikely due to changes of dopamine neuron activity.

**BRG1 in dopamine-targeted neurons is essential for striatal transcriptional responses to repeated social defeat and cocaine treatment.** Stress exposure, as cocaine injections, activates the Extracellular Signal-Regulated Kinase (ERK) pathway, leading to the induction of the expression of those immediate early genes in dopamine targeted regions including the NAc. Such molecular responses are key for the establishment of drugs- and stress-induced long-term behavioral changes[33–37]. We tested whether *c-fos* and *egr1* induction by social defeat could be altered in absence of BRG1. In control mice, an acute social defeat strongly enhanced the expression of both genes within the dorsomedial striatum, the NAc core and the NAc shell. In Brg1[D1Cre] mice, *c-fos* induction was significantly reduced within the dorsomedial striatum and the NAc shell of defeated mice (Fig. 5, upper row), whereas *egr1* induction was fully blocked in any structure tested (striatum, NAc core and shell, Fig. 5, middle row). Similarly, an acute injection of cocaine also induced the expression of *c-fos* gene in control mice, and this induction was absent in mice deprived of BRG1 (Brg1[D1Cre] mice, Supplementary Fig. 7). These defects in immediate early gene expression are unlikely due to an ineffective ERK pathway activation. Indeed, an acute social defeat was able to induce similar levels of ERK1/2 phosphorylation in the NAc of control and Brg1[D1Cre] mice (Fig. 5 lower row). In response to cocaine, we have previously shown that the synergistic action of dopamine and glutamate, through D1 and NMDA receptor (D1R and NMDAR) complexes, triggers a calcium-dependent activation of ERK that launches

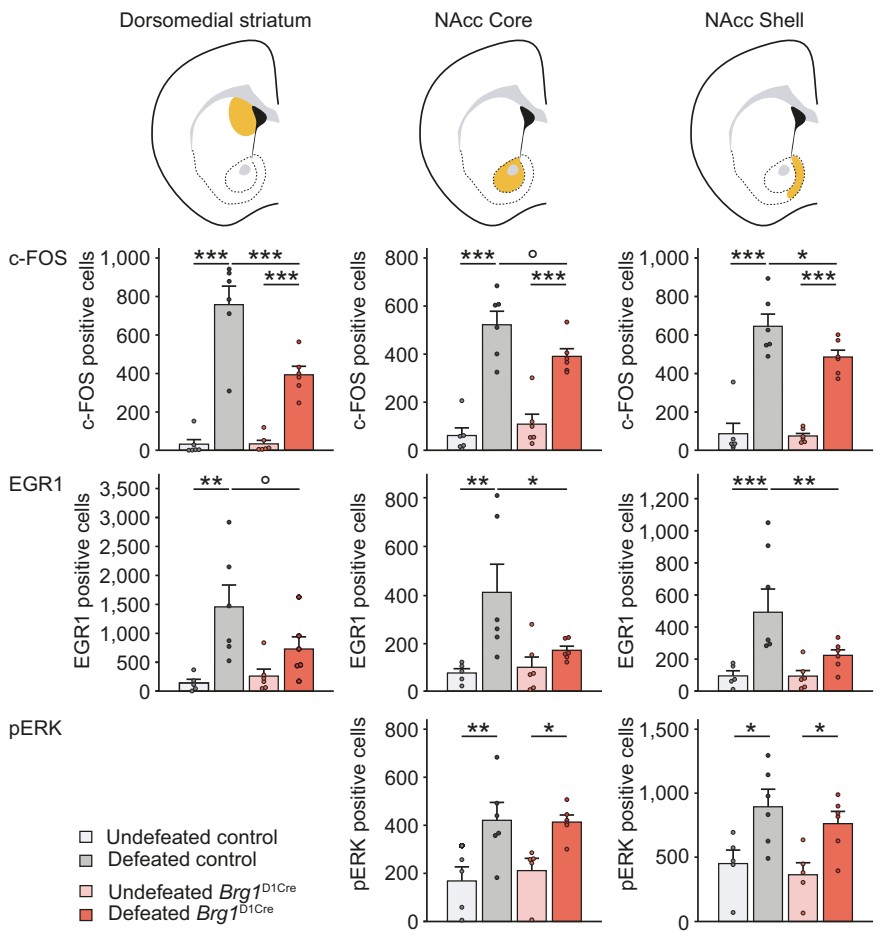

**Fig. 5 Immediate early genes induction of expression and ERK pathway activation in response to a single social defeat in control and _Brg1_[D1Cre] mice.**
First row, _c-fos_ gene induction by an acute social defeat in control and _Brg1_[D1Cre] mice in the dorsomedial striatum (left chart), NAc core (middle chart) and NAc shell (right chart). Locations of the corresponding structures are indicated in orange on the schemes above. Undefeated control $n = 6$, defeated control $n = 6$, undefeated _Brg1_[D1Cre] $n = 6$, defeated _Brg1_[D1Cre] $n = 6$. Two-way ANOVA. Dorsal striatum: main stress effect $F_{(1,20)} = 96.66$; $p < 0.0001$. Interaction genotype x stress $F_{(1,20)} = 10.99$; $p = 0.0035$. Bonferroni correction ***$p < 0.001$. NAc core: main stress effect $F_{(1,20)} = 82.31$; $p < 0.0001$. Interaction genotype x stress $F_{(1,20)} = 4.81$; $p = 0.0403$. Bonferroni correction ***$p < 0.001$, °$p = 0.0679$. NAc shell: main stress effect $F_{(1,20)} = 112.1$; $p < 0.0001$. Bonferroni correction ***$p < 0.0001$, *$p = 0.0455$. Middle row, _egr1_ gene induction by social defeat in controls and _Brg1_[D1Cre] mice in the dorsomedial striatum (CPu, left chart), nucleus accumbens core (middle chart) and shell (right chart). Undefeated control $n = 5$, defeated control $n = 6$, undefeated _Brg1_[D1Cre] $n = 6$, defeated _Brg1_[D1Cre] $n = 6$. Two-way ANOVA. Dorsal striatum: main stress effect $F_{(1,19)} = 14.06$; $p < 0.0014$. Bonferroni correction **$p = 0.0022$, °$p = 0.0766$. NAc core: main stress effect $F_{(1,19)} = 9.44$; $p = 0.0063$. Bonferroni correction **$p = 0.0048$, *$p = 0.032$. NAc shell: main stress effect $F_{(1,19)} = 14.98$; $p = 0.001$. Interaction genotype x stress $F_{(1,19)} = 5.15$; $p = 0.0351$. Bonferroni correction ***$p = 0.0009$, **$p = 0.0075$. Lower row, ERK phosphorylation after an acute social defeat in controls and _Brg1_[D1Cre] mice in the NAc core (left chart) and NAc shell (right chart). Undefeated control $n = 5$, defeated control $n = 6$, undefeated _Brg1_[D1Cre] $n = 5$, defeated _Brg1_[D1Cre] $n = 6$. Two-way ANOVA. NAc core: main stress effect $F_{(1,18)} = 17.96$; $p = 0.0005$. Bonferroni correction **$p = 0.0077$, *$p = 0.0306$. NAc shell: main stress effect $F_{(1,18)} = 15.89$; $p = 0.0009$. Bonferroni correction non defeated _vs_ defeated controls *$p = 0.0163$; _Brg1_[D1Cre] *$p = 0.0316$. Data are represented as mean ± s.e.m. Data are provided in a Source Data file.

subsequent cocaine-evoked epigenetic, transcriptional and behavioral long-term adaptations[38]. To study whether this integration of D1R and NMDAR signaling converging onto ERK was altered in _Brg1_[D1Cre] mice, we used an in vitro model of cultured striatal neurons co-stimulated with a low dose of glutamate together with a D1R agonist, which reproduces the main feature of cocaine-induced signaling[39,40]. As a first step, we measured calcium signal in response to increasing glutamate concentrations, which are mediated by NMDAR stimulation in this model[39]. We did not observe any difference in NMDAR-dependent calcium signal in cultured neurons from control and _Brg1_[D1Cre] mice (Supplementary Fig. 8a-e), suggesting that _Brg1_ deletion did not impact NMDAR-

dependent signaling. By applying the co-stimulation paradigm, we could also show that the D1R-mediated facilitation of calcium entries through NMDAR that is responsible for cocaine-induced ERK activation, was not altered in _Brg1_[D1Cre] mice (Supplementary Fig. 8f, g). Altogether, these results suggest that the absence of BRG1 dampens the induction of expression of immediate early genes without altering signal integration in MSNs.

**Changes of nuclear organization in absence of BRG1.** We next tested whether BRG1 absence could lead to changes of nuclear organization potentially leading to the observed behavioral and

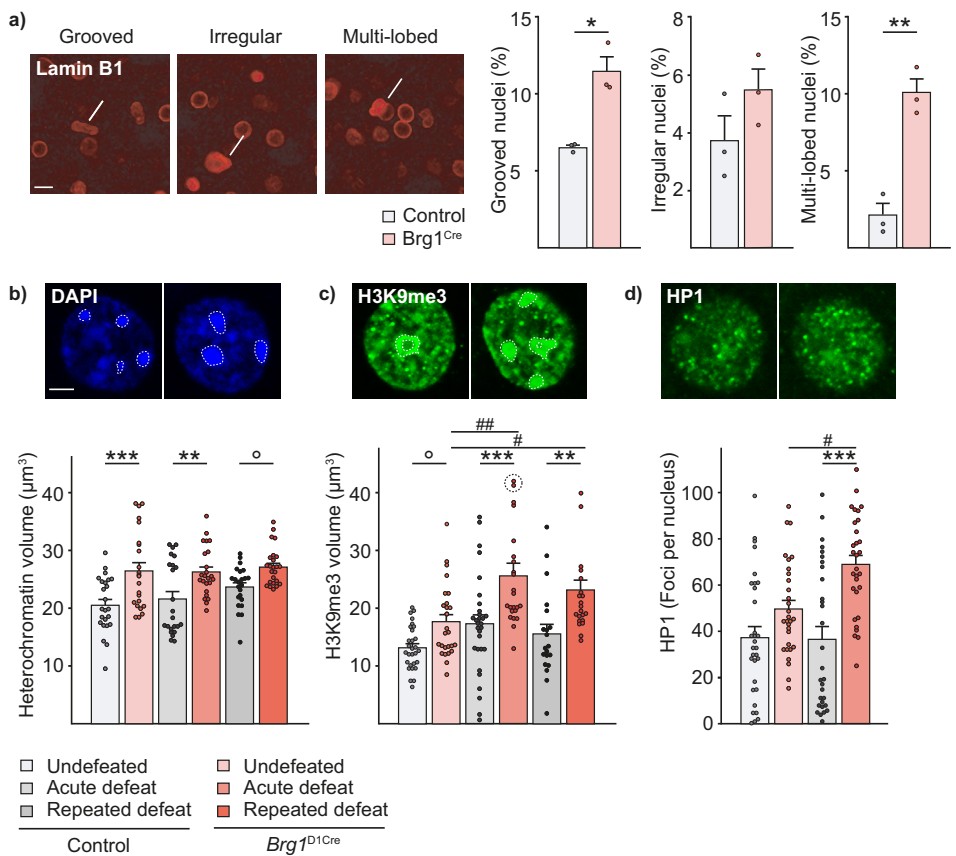

**Fig. 6 Changes of nuclear and chromatin conformation in absence of BRG1. a** Nuclear abnormalities (pointed by yellow bars) in the NAc of control and *Brg1*[D1Cre] mice using Lamin B staining with percentage of grooved, irregular and multilobed nuclei ($n = 3/3$ mice). Scale bar = 20 µm. Two-sided T-test. Grooved nuclei **$p = 0.0066$, multilobed nuclei **$p = 0.0022$. **b** Representative DAPI staining in nuclei from control and *Brg1*[D1Cre] acutely defeated mice with heterochromatin domains circled in dashed white lines. Total volume of heterochromatin in control and *Brg1*[D1Cre] mice undefeated, 1 h after a single defeat or 24 h after repeated social defeat. Scale bar = 5 µm. $n = 24$ nuclei in 4 animals in each condition except for controls with repeated defeat for which $n = 23$ in 4 animals. Main genotype effect $F(1,135) = 30.76$; $p < 0.001$. Two-way ANOVA followed by Bonferroni correction control vs *Brg1*[D1Cre] °$p = 0.0603$, **$p = 0.0052$, ***$p = 0.0003$. **c** Representative H3K9me3 staining in nuclei from control and *Brg1*[D1Cre] acutely defeated mice with quantified domains circled in dashed white lines. Total volume of H3K9me3 staining in control and *Brg1*[D1Cre] mice undefeated (respectively $n = 29$ and $n = 27$ in 5 mice), 1 h after a single defeat (respectively $n = 30$ in 5 mice and $n = 23$ in 4 mice) or 24 h after repeated social defeat (respectively n = 20 and n = 22 in 4 mice). For acutely defeated mutant mice, two points out of scale (52.0 and 52.9 µm³) are pictured circled. Scale bar = 5 µm. Main genotype effect $F(1,145) = 29.31$; $p < 0.0001$, main stress effect $F(2,145) = 8.66$; $p = 0.0003$. Two-way ANOVA followed by Bonferroni correction control vs *Brg1*[D1Cre] °$p = 0.0856$, **$p = 0.0047$, ***$p = 0.0004$; unstressed vs acute stress ##$p = 0.0011$; unstressed vs repeated defeat #$p < 0.0403$. **d** Representative HP1 staining in nuclei from control and *Brg1*[D1Cre] acutely defeated mice. Number of HP1 foci in control and *Brg1*[D1Cre] mice undefeated or 1 h after a single defeat. $n = 30$ nuclei in 5 animals in each condition. Main stress effect $F(1,116) = 3.99$; $p < 0.05$. Main genotype effect $F(1,116) = 23.47$; $p < 0.0001$. Interaction stress x genotype $F(1,116) = 4.78$; $p = 0.0309$. Two-way ANOVA followed by Bonferroni correction control vs *Brg1*[D1Cre] ***$p < 0.0001$; unstressed vs acute stress ##$p = 0.0075$. Data are represented as mean ± s.e.m. Data are provided in a Source Data file.

molecular phenotypes. Recent report has shown that BRG1 may control nuclear shape in non-tumorigenic epithelial cell line due to its impact on chromatin dynamics[41]. We thus asked whether the absence of BRG1 in NAc MSNs could lead to structural changes in the nucleus and to chromatin accessibility in this cell population. Using nuclear envelope Lamin B1 staining and confocal imaging, we found that *Brg1*[D1Cre] mice exhibited a significant increase of abnormally shaped nuclei with an increased number of grooved (6.5% ± 0.2 in controls and 11.4 ± 0.9% in *Brg1*[D1Cre] mice) and multi-lobed nuclei (2.1 ± 0.7% in controls and 10.1 ± 0.9% in *Brg1*[D1Cre] mice) compared to control mice (Fig. 6a). Using DAPI staining, we also assessed the volume of heterochromatin within MSN nuclei of *Brg1*[D1Cre] and control mice, as a proxy of transcriptionally silent regions. Indeed, BRG1 has been associated with transcriptional activation especially in the context of its interaction with nuclear receptors[20]. We found an overall increase of heterochromatin volume in *Brg1*[D1Cre] mice

MSNs in both basal and acute stressed conditions along with a trend for an increase 24 h after repeated defeat (Fig. 6b). These data were consistent with Lysine 9 histone 3 trimethylation (H3K9me3) levels, an epigenetic mark associated with heterochromatin, found in pericentromeric regions. In unstressed conditions, *Brg1*[D1Cre] mice showed a trend toward an increase in H3K9me3 staining volume compared with control mice. This difference was strengthened and reached significance 1 h after a single defeat and 24 h after repeated defeats. After a single defeat, *Brg1*[D1Cre] mice exhibited a strong increase of H3K9me3 levels that tended to stabilize after repeated defeats suggesting that BRG1 has an inhibitory effect on this epigenetic mark upon neuronal activation (Fig. 6c). We furthermore quantified levels of HP1, a protein that is key for heterochromatin packaging, and found again a marked increase in the number of HP1 foci in *Brg1*[D1Cre] mice MSNs 1 h after a single defeat. This was not observed in control mice (Fig. 6d). We also investigated for the

phosphorylation of the serine 10 of histone H3 (H3S10P), a marker of nucleosomal response that that is rapidly activated in response to cocaine and involved in transcriptional activation of immediate early genes[42]. Interestingly, in fibroblasts, it leads to promoter remodeling of some immediate early genes and the onset of their transcription[43]. This process involves MSK1 complexed with BRG1 and precedes SWI/SNF nucleosomes remodeling. Whereas acute social defeat triggered a significant increase of H3S10P positive foci within NAc MSN nuclei of control mice, this was abolished in the absence of BRG1 (Supplementary Fig. 9). Altogether, these data point toward an increased level of condensed chromatin in absence of BRG1.

To investigate the impact of a lack of BRG1 on chromatin organization and genome accessibility in NAc cells, we performed an ATAC-seq (Assay for Transposase-Accessible Chromatin using sequencing) analysis on nuclei purified from the NAc of control and $Brg1^{D1Cre}$ mice. In this assay, genomic regions accessible to an engineered prokaryotic Tn5 transposase are cut, de novo ends, ligated to sequencing adaptors, amplicons corresponding to DNA fragments resulting from two adjacent cut/ligation events, amplified and the resulting DNA libraries, sequenced. This approach allowed us to compare genome-wide differences of accessibility between the two genotypes.

Our ATAC-seq profiles were highly similar to the one obtained from mouse forebrain E15 and P0, as exemplified by read alignment on a portion of chromosome 2 (Methods, Supplementary Fig. 10) and chromosome 11 encompassing the gene Per1, a known target of GR (Fig. 7a; grey box). Profiles of adult NAc and P0 forebrain displayed identical areas of chromatin accessibility, three of which overlapped promoters (see below and Fig. 7a dark blue boxes) that are alternatively used for Per1 transcription. Relative differences in the level of accessibility of those areas between both profiles should indicate tissue and/or developmental stage differences in promoter usages. Several intervals appeared differentially accessible within Per1 gene in absence of BRG1 (see below). Among these, we found one that overlapped the Per1 promoters displaying enhanced accessibility (Fold-change: 1.6, p-value: 4.8E-04) (Fig. 7a, ATAC NAc control lane, red bar). Further intervals displayed altered accessibility of lower statistical confidence, including two matching a GR-bound region defined by a previous ChIPseq study from cultured mouse embryonic fibroblasts 56 (p-value: 1.5 E-02; Fig. 7a, yellow bars). This is also the case for other intervals of other known GR-target genes, as Fkbp5 (chr17:28,486,300–28,486,340; p-value: 9.9E-03), Sgk1 (chr10:21,991,400–21,991,500; p-value: 1.5E-03) and Gilz/tsc22d3 (chrX:140,520,300–140,520,400; p-value 3.0E-03, see file dataset 3 Fig. 7 genes).

We have analyzed reads from individual DNA libraries constructed using the NAc of 7 $Brg1^{D1cre}$ mutants and 6 control littermates (see Methods). Read extremities were counted over the mouse genome (2.7E + 09 bp) binned into 2.7E + 07 intervals of 100 bp. Out of those intervals, 7.1E + 05 (2,4%) harbored a mean of 5 or more read extremities, the threshold value we retained for comparing DNA accessibilities between the mutant and control genomes (Fig. 7b). About 90% of those intervals overlapped with genes (68%) or gene-flanking sequences (22%) (20 kb upstream or downstream from genes) although genes represent 41% of the mouse genome. Interestingly, 46% of the analyzed intervals also overlapped with 99,379 out of 339,142 (29%) cis-Regulatory Elements (cCREs) of the ENCODE registries (including promoters and enhancers in the Per1 region displayed in Fig. 7a) although cCREs represent 3% of the genome intervals. cCREs that are identified for their DNase hypersensitivity (DHSs) and high H3K4me3, H3K27ac and/or CTCF ChIP-seq signals in reference cell lines (Supplementary Fig. 11), indeed display a median size of 285 bp covered by three intervals of 100 bp and, altogether,

overlap some 9.6E + 07 bp and 9.6E + 05 intervals of 100 bp. Finally, while cCREs in intergenic DNA and gene display a 1:2 ratio, most of the cCREs that overlapped the accessible DNA in NAc (94%) mapped within genes.

Fold changes in DNA accessibility (mutant counts/control counts) ranged from 0.2 to 4.7 and associated with p-values ranging from 4.8E-08 to 5.0E-01 (Fig. 7b). Glances at the distribution of the fold-changes in accessibility (mutant counts/control counts) showed a deficit in fold-changes higher than 1 at lowest p-values attesting general decreased accessibility of the $Brg1^{D1cre}$ mutant genome when compared to the control genome (Supplementary Fig. 12). To confirm this, we calculated the ratio of DNA intervals of enhanced accessibility to DNA intervals with repressed accessibility over sliding windows of 1000 intervals, the intervals being ranked by increasing p-values. Ratios were then plotted to the p-values associated with the first window DNA interval (Fig. 7c). A ratio of 0.56, a value indicative of a large excess of DNA intervals with repressed accessibility in the $Brg1^{D1cre}$ mutant relatively to DNA intervals with enhanced accessibility, characterized the lowest p-values. Ratios reached the value of 1.00 as p-values reached the value of 6.0E-03. Differences in the ratios characterizing windows of lower and upper p-values clearly demonstrated a genome-wide effect of the Brg1 inactivation on DNA accessibility.

We performed gene ontology (GO) analyses on the 1,273 gene/DNA intervals identified by the value of 0.60, out of which, 379 overlapped with cCREs mapping within genes or gene-flanking sequences and 394 identified genes or gene-flanking sequences independently of any cCRE. GO analyses of the 736 genes identified by those intervals indicated brain and brain cortex enrichments (p-values of 2.0E-09 and 1.8E-08, respectively) as well as enrichments in the pathways of long-term depression (p-value = 1.2E-03) and morphine addiction (p-value = 1.4E-03) (Supplementary Table 3).

Altogether these data demonstrated a genome-wide effect of Brg1 inactivation on lowering DNA accessibility under basal conditions. Further studies will be required to address whether SWI/SNF could participate in enduring genomic patterning sustaining long-term physiological adaptations to external challenges.

## Discussion

SWI/SNF chromatin remodeler complexes harbor a dozen of protein subunits among which either BRG1 or BRM act as the catalytic ATPase subunit. In this study, we showed that SWI/SNF complexes are essential for some behavioral adaptation to stress exposure through transcriptional responses, a phenomenon essential to adjust behavior to environmental challenges. We have used mouse models with genetic inactivation of the catalytic ATPase subunit that displayed either a lack of BRG1-containing SWI/SNF complexes within all the dopamine-targeted neurons including the deep cortical layers, striatum and NAc (mutant $Brg1^{D1Cre}$) or a lack of BRG1-containing SWI/SNF restricted to NAc neurons (mutant Brg1$^{NAcCre}$). Both models displayed resilience to repeated social defeat meaning they maintained normal sociability. Similar results were observed in mice with a constitutive lack of BRM-containing SWI/SNF complexes (mutant $Brm^{-/-}$), or a lack of any SWI/SNF complexes within the dopaminoceptive neurons combined to a lack of BRM-containing SWI/SNF complexes in other cell types (compound mutant $Brg1^{D1Cre}:Brm^{-/-}$). In addition, $Brg1^{D1Cre}$ mice also showed decreased sensitizing and rewarding responses to cocaine. These behavioral changes appeared independent of a general impairment of fear memory, working memory or motor learning. These enduring alterations of behavioral response to environmental

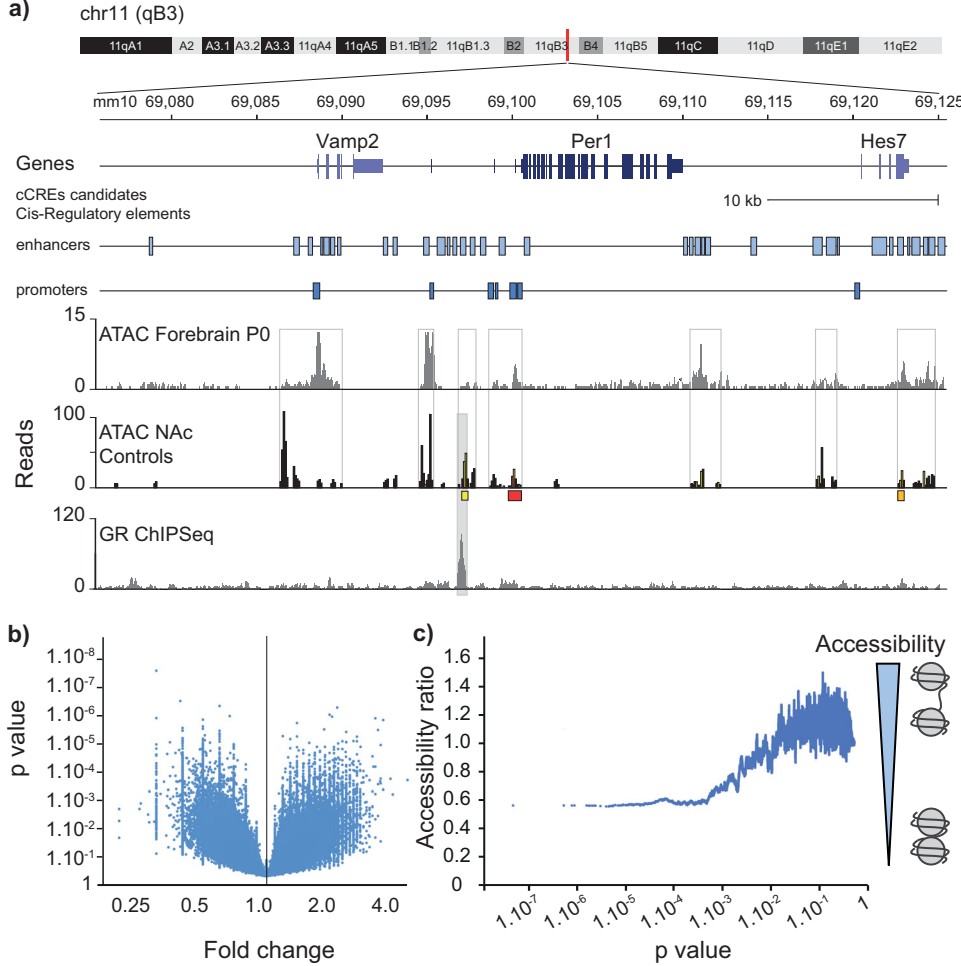

**Fig. 7 Genome accessibility. Accessibility changes in the absence of BRG1. a** Within *Per1* gene locus, regions of accessible DNA in adult NAc match regions of accessible DNA in P0 mouse forebrain taken as a reference profile at UCSC (line boxes). The *Per1* gene, known to bind GR, and its flanking sequences are pictured, as well the ENCODE candidate Cis-Regulatory elements (cCREs) DNA segments (upper lane, light blue boxes for enhancers, lower lane, darker blue boxes for promoters). The profile of read counts from P0 forebrain ATAC-seq experiment is pictured above, the profile of mean counts reads from NAc of control mice (n = 6). An interval with differential accessibility to chromatin between *Brg1*[D1Cre] (n = 7) and control mice (p-value 4.8E-04) is colored in red (position 69,099,900). Intervals with p-value <1.0E-02 and <5.0E-02 are indicated in orange and yellow, respectively. Two-sided Mann-Whitney test. Below, red, orange, and yellow boxes indicate DNA segments containing two adjacent intervals that display differential access with lowest p-value <5.0E-04, <1.0E-02, and <5.0E-02, respectively. Peaks identified throughout a published ChIPseq study directed against GR are shown in the lower lane. **b** Genome accessibility differences between *Brg1*[D1Cre] mutant and control animals. p- values are plotted against fold-changes of accessibility for each of the 7.0E + 05 DNA interval displaying more than 5 read extremities. Two-sided Mann-Whitney test. X- and Y-axes are drawn using log2 and log10 scales, respectively. Values higher than 1 on the X-axis denote enhanced accessibility in *Brg1*[D1Cre] mutant and values lower than 1, repressed accessibility. **c** Ratios of DNA intervals with enhanced accessibility to DNA intervals with repressed accessibility. Ratios were calculated over sliding windows of 1000 intervals, intervals being ranked by increasing p-values, and plotted to the first window p-values. The X- axis is drawn using a log10 scale. We retained the ratio value of 0.60 and corresponding intervals for GO analysis. Data are provided in a Source Data file.

challenges potentially involved a defect of transcriptional response as we observed within the striatum and the NAc and were clearly independent from any change in dopamine neuron activity within the ventral tegmental area. In MSNs of the dorsomedial striatum and of the NAc, a lack of BRG1 reduced the induction of *c-fos* and *egr1* by a single defeat or an acute cocaine injection while the activation of the ERK signaling pathway was unaltered. At the chromatin level, BRG1D1Cre mice showed a blunted induction of Serine 10 phosphorylation of H3 histones after social defeat. This was accompanied by an increase of epigenetic marks associated with heterochromatin such as H3K9me3 and enhanced DNA-packaging protein HP1 levels within MSNs of the NAc, both in basal condition and after social defeats. Finally, ATAC-seq experiments revealed a genome-wide

dependence of chromatin accessibility to BRG1 in the NAc and a bias toward less accessible loci in absence of BRG1.

Repeated social defeat induces enduring anxiety-like behavior and social avoidance that can be normalized by antidepressant treatment[21]. Epigenetic mechanisms are potentially involved to convert repeated acute stress responses into long-term behavioral changes[44]. Studies focusing on DNA and histone modifications suggest that it is indeed the case. Chronic social defeat increases the expression of the DNA methyltransferase 3a in the NAc, the overexpression of which enhances depression-like behavior, while local inhibition of this enzyme has the opposite effect[5]. Perinatal stress in rodents alters DNA methylation in gene promoters such as the one encoding for the arginine vasopressin in the paraventricular nucleus, as well as in genes encoding for GR and

BDNF in the hippocampus and PFC[45,46]. In Human, changes in DNA methylation in *post-mortem* brain samples from suicide completers correlate with severe child abuse[47,48]. At the level of histones, repeated social defeat also leads to changes of acetylation levels in several brain regions including the NAc. Furthermore, local injections of histone deacetylase (HDAC) inhibitors within the NAc or the PFC, ultimately leading to increased acetylation, reverse the social avoidance triggered by repeated defeats[3,49]. Systemic inhibition of histone deacetylation produces antidepressive-like effects comparable to those of regular antidepressants such as fluoxetine[50]. On the opposite, pharmacological activation of HDACs enhances the anxiety- and depressive-like effects of early life stress[51]. Histone methylation has also been shown to be involved in long-term behavioral changes induced by stress and drug-exposure. Both repeated social defeat and cocaine have been shown to downregulate the NAc expression of G9a methyltransferases that catalyze Lysine 9 histone3 dimethylation (H3K9me2), a major transcription repression mark. NAc overexpression of G9a not only made mice resilient to repeated social defeat but also counteracted the additive effect of cocaine exposure potentially through an inhibition of the ERK signaling pathway and of CREB activation[4].

Epigenetic marks participate to the elaboration of transcriptional responses to environmental challenges and stress exposure may, under certain circumstances, push that response toward a pathological path. The adaptation of genome expression requires chromatin remodeling complexes that recognize histone modifications and, through ATP hydrolysis, unwrap, mobilize, exchange or eject nucleosomes, to subsequently recruit the transcriptional apparatus[52]. So far, only few studies investigated the role of chromatin remodeling complexes in behavioral adaptations or in the pathophysiology of psychiatric disorders. Human exome sequencing and genome-wide association studies have linked mutations in genes encoding subunits of the SWI/SNF complex, including *Brm* and *Brg1* to neurodevelopmental disorders such as the Coffin-Siris syndrome[53–55], Nicolaides-Baraitser syndrome[56], autism spectrum disorder[57], and schizophrenia[58]. At the preclinical level, the expression of BRG1 and complex formation with SMAD3 within the NAc has been shown to increase in rats following withdrawal from cocaine self-administration. Intra-accumbal pharmacological inhibition of BRG1 attenuated cocaine-relapse while virally-mediated NAc overexpression enhanced this behavior[17]. Also, BAF53b inactivation in the NAc has been shown to impair cocaine place preference[59]. Our results are complementary since the inactivation of *Brg1* gene in dopaminoceptive neurons including the NAc decreases the locomotor sensitization and the conditioned place preference to cocaine.

Our results extend beyond previous data by demonstrating a role for BRG1 within the NAc in social defeat induced depressive-like and anxious phenotype. Interestingly, *Brm*[-/-] mutant showed a similar absence of social aversion after repeated social defeat, as did the compound mutants bearing the two mutations. These results suggest that both BRG1- and BRM-containing complexes are independently required for the expression of this behavior. Alternatively, it may mean that a minimal activity of SWI/SNF complexes is needed and that the increase of BRM or BRG1 levels respectively observed in *Brg1*[D1Cre] and *Brm*[-/-] mice[60] is not enough to restore a full function. Indeed, evidence suggests some degree of functional redundancy between these two proteins. For example, *Brm*[-/-] mice show an increased anxiety upon repeated social defeats while *Brg1*[D1Cre] do not. One possibility could be that BRG1- and BRM-containing SWI/SNF complexes do play specific complementary functions for some phenotypes, as possibly for behavioral responses to cocaine as Brg1 and compound mutants have a similar phenotype whereas *Brm* gene inactivation has milder consequences. Another possibility could be that

germline inactivation of *Brm* may leave a larger developmental time window for compensatory mechanisms to occur compared with the deletion of *Brg1* in post-mitotic neurons. This hypothesis would fit with the less severe phenotype observed in *Brm*[-/-] mice compared to *Brg1*[D1Cre] and compound mutant mice concerning the sensitization to cocaine. This has however to be moderated by the fact that constitutive inactivation of *Brg1* gene has a much more severe phenotype than that of *Brm* one as embryos die at peri-implantation stage[61]. Another indication of redundancy between BRG1 and BRM lies in the data on locomotor activity, motor learning, and working memory tasks. Indeed, compound mutants virtually lacking functional SWI/SNF complexes in the striatum, NAc and deep cortical layers show clear deficits in locomotor activity, motor learning and working memory relying on these structures while *Brg1*[D1Cre] and *Brm*[-/-] mice did not. This indicates that each protein is individually sufficient to ensure normal motor learning and working memory. Of note, we observed an increase in BRM and BRG1 protein levels in the striata of, *Brg1*[D1Cre], and *Brm*[-/-], respectively. This respective molecular compensation may facilitate the functional compensation in single mutants. Concerning fear memory, while some studies showed an involvement of BAF53b, a member of SWI/SNF complexes[25], none of our models affecting SWI/SNF ATPase units, including the compound mutants, displayed any deficit in either contextual or cued fear memory. This strongly suggests that neither BRM in the entire brain, nor BRG1 in dopaminoceptive neurons are required for fear conditioning. Furthermore, result with the compound mutant show, that no SWI/SNF ATPase activity is necessary in dopaminoceptive neurons for this behavior. These data obviously do not exclude that BRG1 is required for fear memory in other neuronal populations, as contextual and cued fear memory heavily rely on hippocampus and amygdala, two structures in which BRG1 remain expressed in our models.

ISWI, another chromatin remodeler complex also affects stress-related behaviors. Mice susceptible to repeated defeat and patients with depression display, within the NAc, an upregulation of ACF1, a subunit of ISWI. Its overexpression in the same structure, with that of the ISWI ATPase subunit (SMARCA5) made mice more sensitive to social defeat whereas knocking it down made them more resilient[15]. In addition, inactivation of BAZ1B, a subunit of ACF complexes, decreases cocaine self-administration[16]. Increased ACF was associated with altered nucleosome positioning and a repressed transcription of key genes involved in the sensitivity to social defeat[15]. Our findings are similar in that the inactivation of the *Brg1* or *Brm* genes, the two ATPase subunits of the SWI/SNF complexes, made mice resilient to repeated social defeat and decreased behavioral responses to cocaine. However, whereas ACF seems to have a repressive impact on gene transcription, BRG1 has been mainly associated with transcriptional activation[20]. In line with this, we observed in absence of BRG1 a general increase of heterochromatin volume in MSN nuclei, a decreased induction of H3S10P in response to social defeat and, a two-fold excess of loci with decreased chromatin accessibility, thus reinforcing the proposal that BRG1 participates into chromatin accessibility and transcription activation. A related effect has been observed in tumor cells in which the activation of BRG1, due to the inhibition of CDK9 that phosphorylates and inactivates BRG1, prevents its recruitment to heterochromatin and leads to a broad opening of chromatin and the reactivation of transcription of repressed genes[62]. Changes in heterochromatin organization may also be responsible for changes of nucleus shape in MSNs. Indeed, increased heterochromatin may translate into increased lamina-associated domains that directly shape the nuclear envelope[41,63].

The absence of BRG1 leads to a defect of induction of *c-fos* and *egr1* gene expression by social defeat despite an intact cell

signaling transduction (assessed by the MAPK pathway activation). This may partly explain the behavioral phenotype of $Brg1^{D1Cre}$ mice. Indeed, the induction of both immediate early genes has been shown to be crucial for behavioral responses to cocaine[33,35]. Also, c-fos gene expression is under the control of CREB whose inactivation in the NAc has been shown to reduce social avoidance after social defeat[4]. On the other side, egr1 is under the control of Elk1 activation and the systemic injection of a blood-brain-barrier permeable peptide that inhibits ERK-mediated activation of Elk1 has been recently shown to induce resilience to social defeat in mice[64]. In primary cortical neuronal culture, BRG1 has been shown to directly control c-fos gene induction but with a dual action depending on cell activity. Through its interaction with CREST (Calcium Responsive Transactivator), a neuron specific member of SWI/SNF complexes, BRG1 is key for CBP (CREB binding protein) recruitment and for ensuing c-fos gene transcription upon calcium influx[65]. The same mechanism may be at play within the striatum and NAc, potentially accounting for the decreased c-fos gene induction that we observed. In neurons at rest, however, the BRG1-CREST complex rather inhibits c-fos gene transcription through the recruitment of a phospho-Retinoblastoma-HDAC repressor complex[65]. This may explain why we observed a trend toward higher levels of c-FOS in saline injected mice deprived of BRG1. As BRG1 and CREST also bind the egr1 gene promoter, the same mechanism could also explain the lack of egr1 gene induction that we observed. It is of note that here we do not distinguish between the D1- and D2-receptor expressing MSNs and, although beyond the scope of our study, manipulating BRG1 independently in one or the other neuronal population could yield distinct results[66,67].

We previously reported the essential role of GR in dopaminoceptive neurons for behavioral adaptations to stress[7–10]. The fact that SWI/SNF complexes containing either BRM or BRG1 are essential interaction partners of nuclear receptors, among which the GR, led us to address their role in this function. The interaction between GR and BRM or BRG1 is not direct but relies on connecting BAF proteins including BAF250, BAF60a, BAF57, and BAF53a[18–20]. In cell cultures, both BRM and BRG1 potentiate GR transcriptional activity[13,68–70] and studies at a genome scale have shown that, in the fibroblastic 3134 cell line, around 40% of GR-stimulated genes and 11% of GR-repressed genes require the function of BRG1-containing SWI/SNF complexes[71]. Surprisingly, considering the large number of genes supposed to be co-regulated, only a small fraction of GR and BRG1 foci co-localized or were in close vicinity. Obviously, the identification of the genes present within the foci will be of great interest. We followed the evolution of GR/BRG1 distributions after an acute social defeat stress and 24 h after the last defeat of 10 daily successive defeats. An acute stress elicits a burst of glucocorticoids reaching its maximum one hour after its initiation and returning to basal levels 3-4 hours later. These high levels of hormones are most probably responsible for the increased number of GR foci we observed after a single defeat. Surprisingly, acute stress exposure also strongly increased the number of BRG1 foci. This cannot be explained just by a recruitment of BRG1 by GR. Indeed, although the fraction of co-localized GR-BRG1 foci was markedly increased, it still represents only a small fraction of GR foci. Interestingly, 24 h after 10 days of social stress, while the number of GR and BRG1 foci returned to basal levels, the number of foci containing both proteins remained enhanced. This change in nuclear distribution of GR and BRG1 may therefore constitute a remaining molecular trace of stress-exposure. Unfortunately, the lack of antibodies prevented us to perform similar analysis of the relative GR and BRM distributions.

The similarity of the behavioral phenotypes we reported here with the ones previously observed in $GR^{D1Cre}$ mice[7,9], suggests a functional role for the interaction between GR and BRG1/BRM containing SWI/SNF remodeler complexes in the modulation of stress-related behaviors. However, some data in the present study do not match what was previously shown in $GR^{D1Cre}$ mice. Indeed, the inactivation of GR gene in dopaminoceptive neurons leads to a decrease of dopamine neurons spontaneous firing under basal conditions, as well as after repeated social defeats. In contrast, the individual inactivation of either Brg1 or Brm in the same cell population does not. Several hypotheses could explain this difference. GR and BRG1/BRM have functions that are independent from each other, and the control of presynaptic dopamine by GR may require a mechanism that does not involve BRG1 or BRM. This could be exerted at the level of gene expression but also involve non-transcriptional functions of GR such as interferences with signaling cascades or with synaptic activity via plasma membrane-bound GR[72]. An alternate possibility would be that BRM and BRG1 may compensate for each other. This would be in line with our observation that double gene inactivation leads to a decrease of dopamine firing rate and number of spikes in bursts in $Brg1^{D1Cre}Brm^{-/-}$ mice. Further work will be needed to clarify which genes and which functions are under the control of GR-SWI/SNF complexes. However, the present results along with previous work on GR mutants bring one more piece to the understanding of the molecular mechanisms underlying stress-induced behavioral adaptations, and how environmental challenges can pattern genomic organization to shape enduring behavioral changes. Interestingly, the fact that inhibition[17] or genetic inactivation of BRG1 in adulthood, as we showed specifically in the NAc, confers resilience to repeated social defeats and reduces cocaine relapse makes SWI/SNF chromatin remodeler complexes potential targets for mood and addiction therapeutic approaches.

## Methods

**Animal breeding**. Mice (*Mus musculus*) were bred and raised under standard animal housing conditions, at 22 °C, 55% to 65% humidity, with a 12-h light/dark cycle (7 AM/7 PM) and had free access to water and a rodent diet. All experiments were performed in accordance with the European Directive 2010/63/UE and the recommendation 2007/526/EC for care of laboratory animals (decision #7074 2016091623299400). The project received approval from the ethical committee Charles Darwin (Sorbonne Université). Mice were 2 to 4-month-old males. Experiments were carried out during the light phase.

Mice carrying an inactivated Smarca2 gene, generated by homologous recombination, have been previously described[60]. Briefly, an internal exon of the Smarca2 gene (thereafter denominated Brm), corresponding to amino acids 120–276, was disrupted by the insertion of a neomycin cassette. This led to the absence, in $Brm^{-/-}$ mice, of any functional BRM protein able to assemble in SWI/SNF complexes. To inactivate the Smarca4 gene (thereafter denominated Brg1) in dopaminoceptive neurons, we crossed mice carrying a conditional allele of Brg1 gene, sensitive to the Cre recombinase ($Brg1^{L2}$)[73], with the TgYAC-D1aCre mouse line[22]. The $Brg1^{L2}$ allele contains DNA LoxP sites flanking two internal exons. Their excision results in the deletion of the sequence encoding amino acid residues 814 (threonine) to 872 (lysine) and the creation of a frameshift in exon 4 and a stop codon at amino acid position 819. The disrupted gene will therefore encode a C-terminally truncated protein lacking the ATP hydrolysis site.

DNA from mice tails was extracted and analyzed by PCR. For the $Brm^{wt}$ and $Brm^{-}$ alleles the primers were 5'-CCTGAGTCATTTGCTATAGCCTGTG-3' (sense strand), 5'-CTGGACTGCCAGCTGCAGAG-3' (reverse strand) and 5'-CATCGC CTTCTATCGCCTTC-3' (reverse strand in the neomycin cassette). The amplified bands corresponding to the wild-type and the mutated allele are 310 bp and 700 bp in length, respectively. The $Brg1^{+}$ and $Brg1^{L2}$ alleles were detected using primers 5'-GTCATACTTATGTCATAGCC-3' (sense strand) and 5'-GCCTTGTCTCAAA CTGATAAG-3' (reverse strand), primers giving rise to 241 bp and 387 bp DNA segments, respectively. To detect the D1Cre transgene, primers were 5'-GCCTG CATTACCGGTCGATGCAACGA-3' (sense strand) and 5'-GTGGCAGATGG CGCGGCAACACCATT-3' (reverse strand). The amplified band was 700 bp long.

Brg1 and Brm mutant mice were bred on a mixed genetic background derived of 129SVeV and C57BL/6, with a higher contribution of C57BL/6, equivalent to a three-time backcross. $Brg1^{L2/L2}$;TgYAC-D1aCre (thereafter denominated $Brg1^{D1cre}$) mutant mice and control littermates ($Brg1^{L2/L2}$) used in experiments were generated by crossing $Brg1^{L2/L2}$ females with $Brg1^{D1cre}$ males to avoid eventual maternal effect of the mutation. $Brm^{-/-}$:$Brg1^{L2/L2}$ (thereafter denominated $Brm^{-/-}$), compound mutant ($Brm^{-/-}$:$Brg1^{L2/L2}$;TgYAC-D1aCre, thereafter

denominated $Brm^{-/-}$:$Brg1^{D1Cre}$) and control mice ($Brm^{wt/wt}$:Brg1$^{L2/L2}$) were generated using animals issued from $Brm^{wt/-}$:$Brg1^{+/L2}$:TgYACD1aCre x $Brm^{wt/-}$:$Brg1^{+/L2}$ crosses. For locomotor sensitization to cocaine and motor learning experimental individuals were issued from first cousin mice therefore only one control group was used. For all other experiments, respective control littermates were used. $GR^{loxP/loxP}$ and $GR^{D1Cre}$ mice were used for the co-immunoprecipitation experiment[8].

**Stereotaxic injections.** Stereotaxic injections were performed using a stereotaxic frame (Kopf Instruments) under general anesthesia with xylazine/ketamine (10 mg kg$^{-1}$ and 150 mg kg$^{-1}$, respectively). Anatomical coordinates and maps were adjusted from Watson and Paxinos[74]. The injection rate was set at 100 nl min$^{-1}$. $Brg1^{L2/L2}$ mice were injected bilaterally either with a mix of 0.4 μL of pEN-N.AAV.hSyn.Cre.WPRE.hGH (AAV1, gift from James M Wilson, Addgene #105558, titration 1.10$^{13}$ vg/mL) and 0.1 μL of pAAV.hSyn.GFP viruses (AAV1, gift from Bryan Roth, Addgene #50465, titration 7.10$^{12}$ vg/mL), or with 0.1 μL of the later diluted in 4 μl PBS for the control group. The coordinates, set from Bregma, were: AP + 1.5 mm; ML ± 1.2 mm; DV −4.8 mm.

**Histology.** Mice were deeply anaesthetized with pentobarbital (Centravet, France) and transcardially perfused with cold phosphate buffer (PB: 0.1 M Na$_2$HPO$_4$/NaH$_2$PO$_4$, pH 7.4), followed by 4% PFA. Brains were post-fixed overnight in 4% PFA-PBS. Free-floating vibratome sections (30 μm) were rinsed three times with PBS (10 min) and incubated (2 hours) in PBS-BT (PBS-Triton 0.1%, 0.5% BSA,) with 10% normal goat serum (NGS). Sections were incubated overnight (4 °C) in PBS-BT containing 1% NGS with primary antibodies. Sections were then rinsed in PBS and incubated (2 h) with secondary antibodies: anti-rabbit Alexa488 (Goat polyclonal, 1:1000, Invitrogen, Catalog A11008), anti-mouse Alexa488 (Goat polyclonal, 1:1000, Life Technology, Catalog A110029, Lot 1531669), anti-rabbit CY3 (Goat polyclonal, 1:1000, Invitrogen, Catalog A10520),and anti-mouse CY3 (Goat polyclonal, 1:1000, Invitrogen, Catalog A10521),. Sections were then rinsed with PBS (3×10 min) and mounted using Vectashield Mounting Medium with DAPI (Vector Laboratories). Primary antibodies (BRG1, BRM, c-FOS[33], EGR1[75], GR[7], H3K9me3[76], H3S10P[42], HP1[77], Lamin B1[78], NeuN[79], pERK[80]) used are detailed in Supplementary Table 1.

**Co-immunoprecipitation assays and western blots.** Whole striata were dissected and homogenized in 200 μl of radio-immunoprecipitation assay (RIPA) buffer (50 mM Tris- HCl, pH 7.6, 150 mM NaCl, 1% Nonidet P-40, 0.1% SDS, 0.5% sodium deoxycholate, 2 mM EDTA, 0.2 mM sodium orthovanadate and 1X protease inhibitor complete cocktail (EDTA-Free, Roche, 11873580001). Samples were centrifuged for 30 min at 10,000 g and cell lysates were subjected to a second centrifugation for 20 min. A total of 150 μl of lysate from $GR^{loxP/loxP}$ male striata were precleared with 20 μl of protein-A sepharose beads (Sigma, 82506) by a 4 h incubation at 4 °C. Immunoprecipitation was performed by incubating overnight 50 μl of the supernatants each containing the following antibodies: an anti-GR (rabbit polyclonal M20 antibody raised against the N-terminal region, 1:1000 dilution, Santa Cruz Biotechnology, Catalog sc-1004, Lot B0315), an anti-BRG1 (mouse monoclonal, 1:1000, Santa Cruz, sc-17796, clone G-7, Lot H1314) and an anti-BRM (rabbit polyclonal 1:500, Abcam, Catalog ab15597, Lot GR49552-3). The following day, the supernatants were washed three times with 1 ml PBS/Tween and each precipitate was resuspended in 60 μl of 2X Laemmli Buffer. 20 μl of the immunoprecipitated fractions and 20 μg of striata lysates from $GR^{D1Cre}$ and $GR^{loxP/loxP}$ mice were denatured at 95 °C for 5 min. The samples were then separated on a 10% SDS-PAGE gel and transferred onto a nitrocellulose membrane (Bio-Rad, 162-0112). The membranes were incubated in a blocking solution (5% milk in TBS-T: 20 mM Tris-HCl, pH 7.6, 137 mM NaCl, and 0.1% Tween 20) for 1 h at room temperature and then incubated with anti-GR antibody (1:1000) in a 5% BSA in TBS-T overnight at 4 °C. Membranes were washed 3 times with TBS-T, then incubated with peroxidase-conjugated donkey anti-rabbit secondary antibody (1:1000, Jackson Laboratories, Catalog 711-035-152, Lot 94250) diluted in a milk solution 5% BSA in TBS-T for 1 h at room temperature and rinsed 3 times in TBS-T. The proteins were detected using enhanced chemiluminescence (ECL) detection kit (GE HealthCare, RPN2235).

**Social defeat and ensuing sociability and anxiety-like behaviors.** For all tests, animals were daily transferred for habituation in a dedicated room one to two hours before the tests. For each test, chambers of the apparatus were cleaned with alcohol 10% and dried with paper towels between each trial. Behavior was analyzed using Ethovision XT software (Noldus, version 14.0).

*Repeated social defeat and social interaction.* Social defeat was performed as follow. Six-month-old CD1 breeder male mice were screened for their aggressiveness. *Brm* and *Brg1* mutants, as well as their respective control littermates were subjected to 10 consecutive days of social defeat. Each defeat consisted of 5 min of physical interactions between the resident mouse (CD1) and the intruder (experimental mouse). The rotation schedule was set to exclude a repeated defeat by an already encountered resident. Following this, the intruder remained for another 24 h in the resident's home cage that was partitioned in half by a perforated transparent polycarbonate. It allowed visual, auditory, and olfactory communication whilst preventing direct contact. Undefeated mice were handled and rotated daily and housed 2 per cage with the polycarbonate partition separating the cage in 2 halves.

Social interaction was studied 24 h after the last defeat (day 11) in a low luminosity environment (30 lux). Undefeated and defeated mice were introduced for 150 s in a corner of an open-field (40 × 40 × 25 cm) containing in one side an empty perforated polycarbonate box ("no target" condition). Immediately after this, mice were rapidly removed and an unfamiliar CD1 mouse was placed in the box ("target" condition) and mice were re-exposed to the open-field for another 150 s. The time spent in the interaction zone surrounding the polycarbonate box was recorded and used as an index of social interaction.

*Three chambers test.* Sociability was also measured on day 12–14 after the last defeat in a three-chambered box (30 × 15 × 15 cm per compartment) with opening (5×5 cm) in each delimiting wall. The test was divided in 2 phases, each lasting 5 min. During the habituation phase, the challenged mouse was restrained to the central chamber, with the two doorways closed. Doors were then opened, and the challenged individual could either visit the compartment with an unfamiliar adult male mouse (C57BL/6 J) placed in a perforated transparent plastic box or the other compartment containing a similar empty box. The amount of time mice spent in direct interaction with each box was scored.

*Elevated zero-maze.* Elevated zero-maze was performed on day 15–18 after the last defeat. The maze was a circular track (width 7 cm, diameter outer 47 cm) elevated 80 cm from the floor and made of grey polyvinyl chloride (PVC). It was divided in four sections of equal lengths: two opposite open portions and two closed portions with PVC walls, 15 cm in height. At the start, mice were placed in front of a closed portion. Their movement was recorded for 10 min. The latency to enter the open portion, the number of entries and the time spent in the open parts were quantified. A mouse was considered in the open portion when the 4 paws were introduced. The intensity of the light was set to 30 lux in the open parts.

*Dark-light test.* The dark-light test was performed on day 18 after the last defeat. The dark-light box was 45 × 20 × 25 cm, separated into two compartments connected by a central aperture (5 × 5 cm). The dark compartment (black PVC, 15 cm) was covered. The lit compartment (white PVC, 30 cm) was illuminated at an intensity of 30 lux. At the start, individuals were placed in the dark compartment and the latency to enter, and time spent in the lit compartment were measured for 10 min. A mouse was in the lit area when its 4 paws were inside.

**Fear conditioning.** For context conditioning, animals were placed into a conditioning chamber and allowed to habituate for 3 min, followed by three consecutive tone and foot shocks (85 dB 30 s, 0.4 mA shock at the last 2 s) separated by 30 s. Animals remained in the chamber for 1 min after the delivery of the last foot shock. Contextual fear conditioning was measured 24 h after in the same chamber, and freezing was scored for a total of 6 min. Sessions were recorded and freezing was automatically scored. Freezing was defined as complete absence of movement, except for respiration. Cued fear conditioning was measured 24 h after contextual fear conditioning. After changing the context of the chamber (ground, walls, shape, odor, and light color) animals were placed into a new chamber and allowed to habituate for 3 min, followed by tone for 3 min, and freezing was scored for the first 3 min and compared with the last 3 min.

**Motor coordination and learning.** Motor coordination was assessed using a rotarod apparatus (Rotarod/LSI Letica, Biological Instruments). Mice were placed on the rotating rod with an accelerating speed levels (4 to 40 rpm within 2 min) mode. Four trials per day with an intertrial interval of 30 min were performed during 6 consecutive days. The time of the fall was noted for each mouse. The maximum duration of each trial was set to 1 min 30.

**T-maze delayed non-matching to sample task.** Working memory test was performed in a T-Maze with a starting arm and two goal arms[81]. Mice were tested on 10 trials per day, each trial consisting of two runs, a forced run (in which one of the two goal arms is blocked) and a choice run (where both goal arms are open). To obtain a reward, mice were required during the choice run to enter the goal arm not visited during the forced run. The number of correct choices was scored daily, and the task ended when all the mice reached the learning criterion of 7 correct choices out of 10 for 3 consecutive days.

**Behavioral responses to cocaine**

*Spontaneous locomotor activity and drug sensitization.* Locomotor activity was assessed in circular chamber (4,5 cm width, 17 cm external diameter) crossed by four infrared captors (1,5 cm above the base) placed at every 90° (Imetronic, Pessac, France). Locomotor activity was scored automatically when animals interrupted two successive beams, having traveled a quarter of the circular corridor. The spontaneous locomotor activity was measured for 180 min following a first introduction into the locomotor apparatus. For locomotor sensitization to cocaine (Sigma-Aldrich, Saint-Quentin Fallavier, France), mice were habituated to the apparatus for 3 hours for 3 consecutive days and received a saline injection on Days

2 and 3 after 90 min of recording. On days 4 to 7 animals were daily injected with either cocaine (10 mg/kg, *i.p.*) or saline. Following 6 days of withdrawal, all mice received an acute challenge injection of cocaine (10 mg/kg, *i.p.*).

*Conditioned place preference.* The conditioned place preference apparatus consisted of two chambers (10 × 25 × 20 cm) with distinct visual and tactile cues connected by a neutral area. On Day 1 (preconditioning), mice were placed in the neutral area allowed to explore the apparatus freely for 18 min. The time spent in each chamber was measured. On Days 2, 4, and 6, cocaine-paired mice received a cocaine injection (10 mg/kg, *i.p.*) and were confined to one chamber for 20 min. On days 3, 5, and 7, cocaine-paired individuals received saline in the opposite chamber and were also confined for 20 min. Saline-paired animals received saline in both chambers. During the post-conditioning (Day 8), mice (in a drug free state) were allowed to explore both chambers freely for 18 min. The CPP scores were expressed as the increase of time spent in the paired chamber between the post- and the preconditioning sessions.

**In vivo electrophysiological recordings.** Adult male mice (6–12 weeks) were anesthetized with chloral hydrate (8%), 400 mg/kg *i.p.* supplemented as required to maintain optimal anesthesia throughout the experiment and positioned in a stereotaxic frame. A hole was drilled in the skull above midbrain dopaminergic nuclei (coordinates: 3 ± 0.3 mm posterior to bregma, 0.5 ± 0.1 mm [VTA] lateral to the midline)[74]. Recording electrodes were pulled from borosilicate glass capillaries (with outer and inner diameters of 1.50 and 1.17 mm, respectively) with a Narishige electrode puller. The tips were broken under microscope control and filled with 0.5% sodium acetate. Electrodes had tip diameters of 1-2 μm and impedances of 20–50 MΩ. A reference electrode was placed in the subcutaneous tissue. The recording electrodes were lowered vertically through the hole with a micro drive. Electrical signals were amplified by a high-impedance amplifier and monitored with an oscilloscope and an audio monitor. The unit activity was digitized at 25 kHz and stored in Spike2 program. The electrophysiological characteristics of dopamine neurons were analyzed in the active cells encountered when systematically passing the microelectrode in a stereotaxically defined block of brain tissue including the VTA (1). Its margins ranged from −2.9 to −3.5 mm posterior to bregma (AP), 0.3 to 0.6 mediolateral (ML) and 3.9 to 5 mm ventral (DV). Sampling was initiated on the right side and then on the left side. Extracellular identification of dopamine neurons was based on their location as well as on the set of unique electrophysiological properties that distinguish dopamine from non-dopamine neurons in vivo: (i) a typical triphasic action potential with a marked negative deflection; (ii) a long duration (>2.0 ms); (iii) an action potential width from start to negative trough > 1.1 ms; (iv) a slow firing rate (<10 Hz and >1 Hz).

Electrophysiological recordings were analyzed using the R software ([22]). Dopamine cell firing was analyzed with respect to the average firing rate and the percentage of spikes within bursts (%SWB, number of spikes within burst divided by total number of spikes). Bursts were identified as discrete events consisting of a sequence of spikes such that: their onset is defined by two consecutive spikes within an interval <80 ms whenever and they terminate with an inter-spike interval >160 ms. Firing rate and %SWB were measured on successive windows of 60 s, with a 45 s overlapping period on a total period of 300 s.

**Quantification of immediate early genes and pERK in response to cocaine and acute social defeat.** Mice either received an acute injection of 10 mg/kg cocaine or were exposed for 5 min to social defeat. The respective controls received either a saline injection or were similarly handled without aggression. The mice were deeply anesthetized with pentobarbital 1 hour after and went through transcardiac perfusion. The immunostainings were performed on 4 free-floating sections equally spaced along the rostro-caudal axis and encompassing the striatum and the NAc[9]. The images were acquired using a fluorescence microscope equipped with an Apotome module (Zeiss, Axiovert 200 M) and the number of c-FOS and EGR1 positive nuclei were quantified in a blinded manner using Fiji software. Phospho-ERK was quantified the same way, except that the mice were anesthetized, and their brain sampled 10 min after the acute social defeat.

**Quantification of nuclear H3S10P, H3K9me3, HP1, GR and BRG1 distribution in response to social defeat.** Nuclear levels and distribution of H3S10P and HP1 were quantified in unstressed mice and 1 hour after a 5-min social defeat. DAPI-stained heterochromatin, H3K9 Trimethyl, GR and BRG1 were quantified in unstressed mice, 1 hour after a 5-min social defeat (acute condition) or one day after 10 repeated daily social defeats (chronic condition). Undefeated control mice were handled in parallel. The confocal images were acquired with a 63x oil objective, zoom 4, with a 1024 × 1024 pixel resolution (pixel size 0.06 μm) and a Z step of 0.25 μm (Leica-Application-Suite Advanced Fluorescence software (version 2.7.3.9723). Whole nucleus quantification has been performed automatically on 3D stacks in blinded manner using Fiji open access software (version 2.1.0/1.53c). DAPI-stained heterochromatin and H3K9me3 quantification was expressed in volume. The other staining quantifications were expressed in number of nuclear foci.

**Primary striatal cultures.** *Brg1*[D1cre] and control littermate striata were dissected out from E14 mouse embryos and individually cultured. After trypsinization of the tissue (Gibco), for 15 min at room temperature (RT), cells were incubated for 5 min at RT in fetal calf serum complemented with DNase I (Worthington), followed by mechanical dissociation. Cells were then suspended in neurobasal medium supplemented with B27 (Invitrogen), 500 nM L-glutamine, 60 μg/ml penicillin G and 25 μM β-mercaptoethanol (Sigma Aldrich) and plated at a density of 1,000 cells per mm² into 8-μwell plates (BioValley) coated with 50 μg/ml poly-D-lysine (Sigma Aldrich). Cultures were kept at 37 °C in a humidified atmosphere with 5% $CO_2$. Culture medium was changed on the sixth day and live calcium imaging was performed the day after.

**Calcium imaging.** On the seventh day, cultured neurons were loaded for 35 min at RT with a calcium probe Fluo-4 and pluronic F 127 (v/v; 50%; Invitrogen) diluted to 1/500 in a glutamate-free recording medium (129 mM NaCl, 4 mM KCl, 1 mM $MgCl_2$, 2 mM $CaCl_2$, 10 mM Glucose, 10 mM HEPES). Neurons were washed three times in this recording medium before imaging. Acquisitions were performed at 36.2 °C every second, prior to and during treatments with the indicated doses of (R)-( + )-SKF38393 (Sigma Aldrich) and/or L-glutamic acid (Calbiochem), with a spinning disk confocal microscope (Leica) using a 40X oil immersion objective. Fluorescence intensity was analyzed in the soma of all neurons present in each field by using the Fiji software. Viability of the neurons was tested after stimulation with a high dose of glutamate (10 μM), which yields high and sustained calcium responses, at the end of image acquisition. Data are presented as means ± s.e.m. of ΔF/F, ([Ft-F0]/F0 where Ft and F0 are the fluorescence intensities after and before stimulation, respectively). Representative pictures of the ΔF/F signal at the pic of the responses were prepared with MatLab (Version 9.4 R2018a). Calcium imaging experiments were independently performed on 5 littermates. 5 to 23 μ-wells were analyzed for each pharmacological treatment.

**NAc dissection, cell nucleus purification and DNA tagmentation.** ATAC-seq reactions were performed as described in Buenrostro et al. 2013[82], with modifications aimed at optimizing the use of fresh tissue. Animals were sacrificed by cervical dislocation, brains, removed, NAc, dissected from 1 mm slices (between 0.70 mm and 1.70 mm from bregma) and immediately subjected to crush in 1 ml of 25 mM KCl, 5 mM MgCl2, 20 mM Tris pH7.4 (this working solution must be prepared as a 6X stock solution; see below) and 250 mM sucrose (solution A) by using a 2 ml Dounce homogenizer. After centrifugation at 2,000 *g* for 5 min at 2 °C, the crushed tissue was re-suspended into 1 ml composed of 70 μl of the 6X stock solution, 600 μl of solution A and 300 μl of OptiPrep density gradient medium (Sigma). After centrifugation at 10,000 *g* for 10 min at 2 °C, pellets were re-suspended in 200 μl of solution A, 0.1% NP-40 and incubated on ice for 3 min. Permeabilized nuclei were pelleted by centrifugation at 2,000 *g*, for 5 min at 2 °C, re-suspended in 25 μl of 1X transposition buffer containing 2 μl of Tagment DNA Enzyme (Illumina) and incubated at 37 °C for 30 min in a thermocycler. The transposase reaction was stopped by addition of 5 μl of the buffer 0.3 M EDTA and 0.9 M NaCl, 2 μl of SDS 5% and 4 μl of proteinase K [10 mg/ml] and incubated for 1 hour at 37 °C in a thermocycler. The reaction volume was increased to 50 μl and the tagmented DNA was purified using 90 μl of Ampure XP beads (Beckman Coulter) and an elution volume of 40 μl of H2O. 12 μl of the tagmented DNA were then fill gapped and amplified in a 20 μl reaction volume using 1 μM dual-index PCR primers[82] and 1X Phusion PCR Master Mix in the following conditions: 65 °C for 10 min, 72 °C for 2 min followed by 18 cycles of 98 °C for 20 s, 63 °C for 30 s and 72 °C for 60 s. The volume of the PCR products was increased to 50 μl and added with 32 μl of Ampure XP beads. Supernatants were then added with 58 μl of Ampure XP beads and PCR products, finally eluted in 40 μl of H2O. We constructed individual (one library/animal) ATAC-seq libraries from the NAc of nine *Brg1*[D1cre] mutant and eight control animals. Libraries were paired-end sequenced (37 and 38 bp) on an Illumina Nextseq instrument at the TGML (Marseille, France) and i2bc (Gif sur Yvette, France) platforms.

**Primary data processing, validation, and data analyses.** Reads were aligned to the mouse mm10 reference genome, mitochondrial and nuclear PCR duplicate reads, removed, and proper pairs (excluding those mapped to unmapped contigs), selected, by using Bowtie2 (Version 2.4.2), Samtools (Version 1.11) and Picard Tools (Version 2.18.2) of the usegalaxy platform (https://usegalaxy.org). Depending on the library, we recovered 3.8E + 6 to 6.9E + 7 nuclear unique reads. Unique nuclear reads displayed subnucleosomal (≤140 bp), nucleosomal (141-280 bp) or poly-nucleosomal (≥281 bp) characteristics lengths. Ratios of reads of sub-nucleosomal, nucleosomal and poly-nucleosomal lengths varied between animals indicating the extreme sensitivity of the nuclear purification/ATAC-seq protocol when working with animal tissues (Supplementary Table 2). As we wanted to compare data from libraries displaying different ratios of subnucleosomal, nucleosomal and poly-nucleosomal read lengths, we first compared the qualitative and quantitative information provided by those three subsets of reads relatively to DNA accessibility. Nucleotides at read extremities indicate sites of DNA accessibility. We therefore counted read extremities that mapped within a 6-kb region of chromosome 2 selected for its large number of reads allowing deep analysis (chr2:98,664,000-98,670,000) for each read subset. The read extremities of the

library corresponding to Mutant Brg1[D1Cre] Replicate 4/18 clearly identified three regions of 70 bp (chr2: 98,664,030-98,664,100), 200 bp (chr2: 98,665,020-98,665,230) and 1 kb (chr2: 98,666,630-98,667,630) (Supplementary Fig. 10). Those regions precisely matched the regions that had previously been identified by the same type of assay, displayed at the UCSC browser, and taken as the reference assay. That assay used nuclei from the forebrain of P0 mice thereby demonstrating the quality of our data and similar DNA accessibilities in adult NAc and P0 forebrain. Only the extremities of the read subsets of subnucleosomal and nucleosomal lengths (18,585 and 10,263, respectively) identified the three regions, the extremities of the poly-nucleosomal read subset barely identified the 70-bp and 200-bp regions and identified only part of the 1 kb one although counting a similar number of read extremities (10,411). (Supplementary Fig. 10b). The short length of the 70-bp region and the DNA length separating the 70 bp, 200 bp and 1 kb regions must account for this: the two extremities of poly-nucleosomal fragments indeed cannot map within a region shorter than 281 bp and no extremity of poly-nucleosomal fragments can map within a region shorter than 281 bp if this one is too far away from another accessible region. Altogether our data showed that the subnucleosomal, nucleosomal and poly-nucleosomal read subsets produced similar but slightly different information regarding DNA accessibility. This in turn implies that library comparison must be performed on read subsets of subnucleosomal, or nucleosomal lengths rather than on whole datasets of different subset ratios. For our analyses we selected the subset of nuclear unique reads of subnucleosomal length from each library. Read extremities were counted over the mouse genome (2.7E + 09 bp) binned into 2.7E + 07 DNA intervals of 100 bp. Six control and seven Brg1[D1cre] mutant sub-libraries were kept for comparison; two Brg1[D1cre] mutant and two control sub-libraries were excluded because of their low count numbers (9.6E + 05 to 2.2E + 06 reads) compared to the others (1.7E + 07 to 3.3E + 07 reads) and, as a consequence, a lack of counts in a large fraction of the genome intervals. Counts were normalized using a DESeq procedure. Gene ontology analyses were performed by using the DAVID platform (https://david.ncifcrf.gov).

**Statistics and reproducibility**. Statistics have been analyzed using PRISM software (version 7.2). Data are presented as means ± s.e.m. Statistical analysis was carried out using either one-way or two-way ANOVA followed by Bonferroni's multiple comparisons. Unpaired Student's t-test were used for the nuclear shape quantification, for the frequency of dopamine firing rates and for the calcium imaging analysis. Mann-Whitney tests were used for the spike in burst analysis, for compound mutant anxiety test and ATAC-seq analyses.

GR-SWI/SNF co-immunoprecipitations, acute GR-BRG1 co-localizations, NeuN staining quantifications, social avoidance and anxiety tasks following repeated social defeats in Brg1[D1Cre] and Brm[-/-], electrophysiological recordings of dopamine neurons, Ca[2+] imaging, quantification of heterochromatin amount in DAPI stainings, have been performed in several independent experiments yielding the same results. DAPI stainings have been confirmed with H3K9me3 and HP1 stainings. Differences in induction of c-fos gene expression between Brg1[D1Cre], and control mice have been examined upon two different conditions (cocaine and stress responses) giving similar results.

Cocaine sensitization has been done once. The effects of the mutations were clear and coherent with the results obtained with Brg1[D1Cre] mice in conditioned place preference to cocaine as well as in line with experiments showing decreased sensitization to morphine in that model. Rotarod data come from a single cohort. The results show very low variability and match the locomotor activity measured in Supplementary Fig. 3c. Social aversion following repeated defeats in the compound mutants and in NAc inactivated model result from a single experiment, done upon reviewers' request. The data follow our findings on Brg1[D1Cre] and Brm[-/-] models. Finally, fear memory and T-maze in the Brm[-/-] and compound mutants have been done once but showed very little variability with a cohort size between 9 and 11 animals.

We measured once the nuclear shape abnormalities as well as the BRM and BRG1 levels in respective mutants as these results were in line with already published data (Imbalzano et al. 2013, and Reyes et al.1998, respectively). ATACseq sequencing experiments were performed within standard criteria for this approach. Each sample used 6 to 7 biological replicates of NAc. The regions of accessible chromatin in each biological replicate match the region of accessible chromatin identified in similar experiments. Furthermore, the differentially accessible regions between control and mutant NAcc are enriched in brain specific genes.

**Reporting summary**. Further information on research design is available in the Nature Research Reporting Summary linked to this article.

## Data availability

The ATAC-Seq data generated in this study have been deposited in the SRA database under accession code PRJNA751751. For analyses, we used publicly available datasets, including mouse mm10 reference genome (https://www.ncbi.nlm.nih.gov/assembly/GCF_000001635.20/), GR ChIPseq study from cultured mouse embryonic fibroblasts 56 (https://www.ncbi.nlm.nih.gov/geo/query/acc.cgi?acc=GSE126655), and ATAC-seq profiles obtained from mouse forebrain E15 (https://www.encodeproject.org/experiments/ENCSR976LWP/) and P0 (https://www.encodeproject.org/experiments/

ENCSR310MLB/). Raw Data for Figs. 1–7 and Supplementary Figs 1–12 have been provided in the Source Data file'. There is no restriction on data availability. Source data are provided with this paper.

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

## Acknowledgements

Authors wish to thank the IBPS imaging and animal facilities, especially J.F. Gilles and F. Machulka, for technical support and BioArt platform for discussion. Authors acknowledge the high-throughput sequencing facility of I2BC (Gif sur Yvette, France) and TAGC (Marseille, France) for their sequencing and bioinformatics expertise, and L. Marion-Poll

for her advice on nuclei preparations. This work was supported by a Sorbonne Université grant (Emergence to FT), the Labex BioPsy (to FT and SP), the Foundation for Medical Research (FRM Equipe grant DEQ20140329552 to FT), the Institut National du Cancer (TABAC grant to PF, JB and FT), and the Agence Nationale de la Recherche (ANR3053NEUR3143830 to FT, ANR-14-CE35-0029-01 to SP, and ANR-15-CE16-0017 to JB and PV).

## Author contributions

A.Z.C.B., A.C.C., C.V., S.K., R.D.C., E.S.J., S.B., F.M., LA, and SP performed experiments; A.Z., C.B., C.V., R.D.C., E.S.J., F.M., P.V., J.B., L.A., F.T., and S.P. analyzed the data; F.M., P.V., P.F., J.B., F.T., and S.P. supervised the experiments; F.T. and S.P. designed the study; A.Z., L.A., F.T., and S.P. wrote the manuscript.

## Competing interests

The authors declare no competing interests.
