## [Peer Review File · Nature Communications]

SWI/SNF chromatin remodeler complex within the reward pathway is required for behavioral adaptations to stressReviewers' comments:

Reviewer #1 (Remarks to the Author):

In this manuscript, the authors investigate the role of the SWI/SNF ATPase subunits (BRG1 and BRM) in response to social defeat stress and cocaine. To do this they make use of a viable germ line BRM knockout mutant mouse and a conditional depletion of BRG1 in neurons expressing dopamine receptors (TgYAC-D1aCre). These mutants were tested in many different behaviour assays in response to social defeat and cocaine. In response to social defeat both mutants show altered behaviour compared to the controls in social interaction. Only the conditional BRG1 mutant shows altered behaviour response to social defeat in the elevated O maze, 3 chamber test, and dark light box. The BRM mutants were not tested in some of the assays.

BRG1 conditional mutants show a normal acute behavioural response to cocaine, but have a dramatically altered response after chronic cocaine exposure. BRM germline knockout mice do not show an altered behaviour response to cocaine. Interestingly, the double mutant (BRG1 conditional + BRM germline) do not respond normally to acute or chronic cocaine exposure.

The conditional BRG1 knockouts were tested in several learning and memory assays – fear conditioning, motor learning rotorod, working memory. They were normal in these assays. BRM mutants were only tested in the motor learning assay. Interestingly, both BRG1 conditional and BRM KO showed normal motor learning – while the double mutant showed no motor learning ability. To compliment the behaviour tests electrophysiology was performed. BRG1 conditional and BRM KO mice were similar to controls, but the double mutant showed decreased frequency and spike in burst. At the cellular level – the authors measured abnormalities in nuclear morphology and attempted to quantify heterochromatin by measuring the area of intense DAPI staining. BRG1 conditional mutants showed an increase in both while BRM KO mice were not examined. In addition, they looked at numbers of cells expressing cFos, EGR1. The number of cells expressing these IEGs increases after social defeat and cocaine, but the increase is not as much in BRG1 conditional mutants. Upstream signaling components that activate IEGs (phospho ERK, glutamate mediated calcium responses) were normal. H3S10phos was increased in response to social defeat, but not in conditional BRG1 mutants. This is an interesting paper that definitely gives some unique insight into the role of SWI/SNF complexes in long term drug and stress responses, but I have several concerns regarding the relative contributions of BRG1 vs BRM, and with the chromatin analysis.

1. Any conclusions regarding compensation and potential redundant or unique characteristics of BRG1 and BRM are not well founded. In some assays (like social interaction after social defeat) the BRG1 and BRM mutants behave similar. In others, (cocaine response) they don't. BRG1 is only deleted in neurons expressing dopamine receptors while the BRM is a germline mutant. Any differences between the two mutants could be due to the different nature of the mutations rather than the difference between the function of the two genes.

2. On the other hand, for some assays (rotorod – electrophysiology) both mutants were tested alongside a double mutant. These experiments are very informative because they show that there is likely compensation happening as the double mutants display a more severe phenotype. This means that there are likely different thresholds of SWI/SNF activity that are required for different aspects of neuron function. This might explain some of the results in this study that contradict the literature. For example SWI/SNF complexes have been implicated in memory in mouse fear conditioning and in flies (I believe that these studies are not even cited here and they should be). The lack of memory defects observed here might be due to overall different levels of SWI/SNF activity compared to other studies that target different components of the complex. This aspect of the manuscript would be greatly improved if all assays could be done on all three mutant conditions. One could assume that the two single mutants represent conditions with partial SWI/SNF activity, while the double mutant has complete loss of SWI/SNF activity. However, it would be best if the level of partial activity could be quantified in the BRG1 conditional and BRM KO. In that way the authors would be able to compare a defined allelic series representing different known levels of SWI/SNF activity. This would allow to make more robust conclusions about the role of SWI/SNF complexes in the different behaviors and cellular responses.

3. The paper relies heavily on quantifying antibody labeling. It is important to back this up using a different method, for example qPCR, RNAseq, ChIP (for histone marks). For example – for H3S10phos it would be much more informative if they could show by ChIP that this mark was not properly induced at IEGs that are differentially induced in BRG1 mutants. That said, it is not clear why H3S10 was investigated. In the context of SWI/SNF function it would be more logical to study regions of open chromatin, or nucleosome positioning. There are also several other histone modifications known to be associated with IEG induction so why not look at those?

4. Evidence for antibody specificity should be provided for all antibodies. Ideally they should show lack of signal from a mutant tissue or cell line. An alternative would be to use blocking peptides. If this information is already available in the literature it should be cited.

5. The increase in intense regions of DAPI staining (representing heterochromatin) is interesting but does not confirm increased heterochromatin. How was this quantification done? It would need to be done on 3D images of sufficient resolution in the z plane to get an accurate measure of heterochromatin volume. There is not a clear explanation as to how the heterochromatin surface was measured or how the DAPI threshold intensity set to make those measurements? Were experimenters blinded when making these measurements. Alternate methods should be used to show expanding heterochromatin. Preferably ChIP-seq using known markers of heterochromatin. Alternately, immunostainings using known markers for heterochromatin.

6. In general the connection between SWI/SNF, GR, and IEG expression is weak. This could be strengthened by more molecular data (ChIP) examining the interaction of these factors at spec

Reviewer #2 (Remarks to the Author):

The manuscript is a sound study, based on substantial amount of interesting and novel data to reach the conclusion, i.e. the chromatin remodeler SWI-SNF are important epigenetic player in chronic social defeat (CSD) stress induced maladaptive changes in reward circuitry in depression. The authors demonstrate how stress exposure induces stable reorganization of the transcription factor Glucocorticoid Receptor (GR) and chromatin remodeler sub unit BRG1, in the nuclei of striatal medium spiny neurons (MSNs). They show that SWI-SNF complex subunits BRG1 and BRM are essential in dopaminergic neurons for the development of depression-like phenotype i.e. social avoidance and anxiety following repeated CSD. Lack of BRG1 and BRM in mutant mice was the reason for the attenuated responses to cocaine, as shown in locomotor sensitization and conditioned place preference (CPP) paradigms.

Behavioural changes or phenotype in animals that are exposed to chronic stressors and/or psychostimulants such as cocaine have been shown to be owing to the change in dopamine neurons. However, the absence of BRG1 or BRM in these models does not alter this neuronal population. They suggest that important action of BRG1 was through transcriptional response in striatal MSNs; they found that the induction of immediate early genes following Serine 10 phosphorylation on histone H3 on their promoters were significantly down on stress exposure resulting in heterochromatinization, thus affecting transcription.

However, I have following criticisms/suggestions:

1. The authors have shown that Brg1^{D1Cre} mice do not show any abnormalities in learning and memory. However, they have not talked about the constitutive Brm knockout mice in this context. It would be interesting to know if any defects of learning and memory exist in these mice.

2. Under the results section titled 'Brg1 gene ablation in dopamine-innervated areas prevents behavioural adaptation induced by repeated social defeat', the authors should highlight the

differences in anxiety-like behaviour observed between the Brg and Brm knockout mice. The authors might also consider revising the statement 'Indeed, while repeated defeats significantly increased anxiety levels, no difference was observed between Brm^{-/-} mice and their control littermates', since it does convey the result very clearly. Also, a possible explanation for this should be discussed.

3. The conditional place preference test was performed with the Brg1D1Cre mice, but not with Brm KO mice. The authors should either do the test or at least discuss its possible outcome, considering that Brg1 and Brm might have complementary roles.

4. The discussion should include a mention about the inhibitors of Brm and Brg1. Since a knockout of these genes confers stress resilience and reduced cocaine sensitisation, the possibility of using their inhibitors as therapeutic agents for mood disorders and addiction can be discussed.

5. Overall, the manuscript is well written, but has some typos and grammatical errors which require proofreading.

Reviewer #3 (Remarks to the Author):

In this study, the authors examined the function of BRG1 and BRM, two ATPase subunits of the SWI/SNF chromatin complex, in stress behaviors within the reward pathway. Though chromatin remodeling was reportedly implicated in stress and addiction in the past, the role of BRG1 and BRM in this process is still elusive. The authors have carried out extensive behavioral, morphological, electrophysiological approaches, in addition to some other experimental assays, to illustrate the functional role of BRM and BRG1 in stress response. The data are interesting and the manuscript has an in-depth discussion. One limiting factor of this study is there is no direct examination of chromatin modification change at stress relevant genes.

In addition, here are a few other comments:

1. The study is mainly based upon a conditional BRG1 knockout and a conventional BRM mutant mouse line. Do the two molecules have compensatory expression change when the other's transcription is decreased? I also feel the study on the BRM mutant mice does not add in much more insight. What are the differences between the two molecules? Do they function differently?
2. It seems BRG1 was solely knocked out in D1-MSNs, but not D2-cells, which have different roles in stress and addiction. The author should therefore specify their findings are to D1-MSNs only throughout the manuscript. Does this cre-loxp KO also exist in other brain regions? The authors should carry out a rescue assay to make the findings more conclusive.
3. It also appears the mice in this study were not congenic. Depends on the strain, the mice may have a mixture of J129, C57bl6, and or CD1. Have the authors tried backcrossing? The authors should examine and report the exact genetic background of each transgenic mouse line they used.

Response to reviewer #1:

“In this manuscript, the authors investigate the role of the SWI/SNF ATPase subunits (BRG1 and BRM) in response to social defeat stress and cocaine. To do this they make use of a viable germ line BRM knockout mutant mouse and a conditional depletion of BRG1 in neurons expressing dopamine receptors (TgYAC-D1aCre). These mutants were tested in many different behaviour assays in response to social defeat and cocaine. In response to social defeat both mutants show altered behaviour compared to the controls in social interaction. Only the conditional BRG1 mutant shows altered behaviour response to social defeat in the elevated O maze, 3 chamber test, and dark light box. The BRM mutants were not tested in some of the assays.

BRG1 conditional mutants show a normal acute behavioural response to cocaine but have a dramatically altered response after chronic cocaine exposure. BRM germline knockout mice do not show an altered behaviour response to cocaine. Interestingly, the double mutant (BRG1 conditional + BRM germline) do not respond normally to acute or chronic cocaine exposure. The conditional BRG1 knockouts were tested in several learning and memory assays – fear conditioning, motor learning rotorod, working memory. They were normal in these assays. BRM mutants were only tested in the motor learning assay. Interestingly, both BRG1 conditional and BRM Kos showed normal motor learning – while the double mutant showed no motor learning ability.

To compliment the behaviour tests electrophysiology was performed. BRG1 conditional and BRM KO mice were similar to controls, but the double mutant showed decreased frequency and spike in burst.

At the cellular level – the authors measured abnormalities in nuclear morphology and attempted to quantify heterochromatin by measuring the area of intense DAPI staining. BRG1 conditional mutants showed an increase in both while BRM KO mice were not examined. In addition, they looked at numbers of cells expressing cFos, EGR1. The number of cells expressing these IEGs increases after social defeat and cocaine, but the increase is not as much in BRG1 conditional mutants. Upstream signaling components that activate IEGs (phospho ERK, glutamate mediated calcium responses) were normal. H3S10phos was increased in response to social defeat, but not in conditional BRG1 mutants.

This is an interesting paper that definitely gives some unique insight into the role of SWI/SNF complexes in long term drug and stress responses, but I have several concerns regarding the relative contributions of BRG1 vs BRM, and with the chromatin analysis.”

We thank Reviewer #1 for his comments and concerns. As described below in our point-by-point answer, we now provide additional experiments aimed at studying the relative contributions of BRG1 and BRM in the tested behaviors. In addition, we demonstrate that Brg1 gene inactivation in the NAc is sufficient to reproduce Brg1^{D1Cre} phenotype following repeated social defeats. As requested, we have included additional immunostainings and quantifications of heterochromatin levels in Brg1^{D1Cre} mice compared to their controls. Finally, we provide a genome-wide analysis of chromatin/DNA accessibility in Brg1^{D1Cre} mutants and their controls using the ATAC-seq technology.

Response to specific points:

Points 1 and 2. *”1. Any conclusions regarding compensation and potential redundant or unique characteristics of BRG1 and BRM are not well founded. In some assays (like social interaction after social defeat) the BRG1 and BRM mutants behave similar. In others, (cocaine response) they don't. BRG1 is only deleted in neurons expressing dopamine receptors while the BRM is a germline mutant. Any differences between the two mutants could be due to the*

different nature of the mutations rather than the difference between the function of the two genes.

2. On the other hand, for some assays (rotarod – electrophysiology) both mutants were tested alongside a double mutant. These experiments are very informative because they show that there is likely compensation happening as the double mutants display a more severe phenotype. This means that there is likely different thresholds of SWI/SNF activity that are required for different aspects of neuron function. This might explain some of the results in this study that contradict the literature. For example SWI/SNF complexes have been implicated in memory in mouse fear conditioning and in flies (I believe that these studies are not even cited here and they should be). The lack of memory defects observed here might be due to overall different levels of SWI/SNF activity compared to other studies that target different components of the complex. This aspect of the manuscript would be greatly improved if all assays could be done on all three mutant conditions. One could assume that the two single mutants represent conditions with partial SWI/SNF best if the level of partial activity could be quantified in the BRG1 conditional and BRM KO. In that way the authors would be able to compare a defined allelic series representing different known levels of SWI/SNF activity. This would allow to make more robust conclusions about the role of SWI/SNF complexes in the different behaviors and cellular responses.

Response to points 1 and 2:

The new discussion on whether BRM- and BRG1-containing SWI/SNF complexes have redundant or unique complementary functions is enriched by the fact that we have now performed most behavioral tests with each single and the compound mutants.

We wished to address behaviors related to dopamine innervated brain regions - we generated a *Brg1* gene inactivation targeted to this neuronal population - and we therefore focussed our analyses and discussion on dopamine-dependent behaviors that include behavioral changes after repeated social defeat, locomotor sensitization to cocaine, motor learning and working memory. As pointed by reviewer #1, given the different nature of both single mutations it is not pertinent to address phenotypes that do not engage dopaminergic neurons. We will come back at the end of this answer to this point.

Concerning motor learning, our initial data suggested that BRG1 and BRM have redundant functions since the ablation of each single protein did not have any consequences whereas the compound mutant did not learn the rotarod task. Concerning locomotor sensitization to cocaine, our data suggested that BRG1 has a unique function since its absence in dopaminergic neurons has the same consequences than the absence of both BRG1 and BRM proteins, whereas BRM absence has no, or mild effect.

We completed the behavioral study of *Brm*^{-/-} and compound mutants, including repeated social defeats, working memory, locomotion, and fear-conditioning. Our data suggest that for working memory and locomotion, motor learning, BRG1 and BRM have redundant functions. For repeated social defeats, we observed a different situation. The compound mutant loses social aversion in a way similar to that of each single mutant. This could either be explained by both proteins having unique complementary functions or by the fact that a threshold of SWI/SNF activity is required and cannot be fulfilled by each individual gene in dopaminergic neurons. These new data have been included in figures S3 and S4 and are discussed on page 28.

At the physiological level, the fact that *Brm*^{-/-} and *Brg1*^{D1Cre} mutants have firing and bursting activities of VTA dopamine neurons similar to that of control animals, whereas compound mutants show a marked decrease in activity, suggests also functional redundancy between BRG1 and BRM.

At the molecular level, compensatory expression in single mutants of genes sharing overlapping functions can be an indication of functional redundancy, as for instance in the case

of genes encoding the *creb*, *crem* and *atf1* transcription factors (Mantamadiotis et al. 2002, *Nature Genetics*). Such a compensatory expression applies for BRM and BRG1. The constitutive inactivation of *Brm* results in an increase of BRG1 levels in the brain (Reyes et al. 1998, cited in the manuscript). We thus quantified BRM and BRG1 in the striatal neurons of *Brg1* and *Brm* mutants, respectively. As expected, for *Brm* mutants, we observed an increase by 31% of BRG1 protein levels. We show that, conversely, the inactivation of *Brg1* leads to a significant increase of BRM protein levels of 52% (See graphs below). These adaptations in BRG1 and BRM protein levels may compensate for the absence of the other protein and thereby participate to functional redundancy observed in some behaviors. This could explain for instance the motor learning data for which only compound mutants showed a deficit. These data are now cited in the results section page 17, as data not shown, and mentioned in the discussion (page 29).

Concerning the different nature of the mutation, a marked difference is that the *Brm* gene is absent from the beginning of the formation of the embryo whereas the *Brg1* one is inactivated in post-mitotic neurons of circumscribed brain regions. This could lead either to developmental adaptations in *Brm*^{-/-} mice, or to interferences with phenotypes associated with *Brm* gene functions in other cell types than dopaminergic neurons. This point is now addressed in the discussion section page 29. See also our answer to point 2 below.

Reviewer #1 mentions that some of our results contradict the literature on the role of SWI/SNF complexes on memory defects, citing fear memory studied in mice and flies on Brg1 Associated Factors. We rather think our results largely fit with previous work and add some interesting knowledge on the role of SWI/SNF complexes in this behavior. These studies are now cited. We performed fear-conditioning experiments to exclude that a defect in fear-memory would explain our observations on repeated social defeat (see page 19). Following suggestion of Reviewer #1, we now performed and compared all three mutants in fear-conditioning. As none of our three models have a defect in fear memory, either contextual or cued, and as we directly targeted catalytic members of SWI/SNF complexes (ATPase subunits), our results demonstrate that neither BRM in the entire brain, nor BRG1 in dopaminergic neurons are required for fear conditioning. Furthermore, the compound mutant show, that no SWI/SNF ATPase activity is necessary in dopaminergic neurons for this behavior. These data obviously do not exclude that BRG1 is required for fear memory in other neuronal populations. Since contextual and cued fear memory heavily rely on hippocampus and amygdala, two structures in which BRG1 remain expressed in our models, it would be interesting to address this question in another study. Those data have been incorporated and discussed within the new version of our manuscript.

Point 3. "The paper relies heavily on quantifying antibody labeling. It is important to back this up using a different method, for example qPCR, RNAseq, ChIP (for histone marks). For example – for *brgphos* it would be much more informative if they could show by ChIP that this mark was not properly induced at IEGs that are differentially induced in BRG1 mutants. That

said, *It is not clear why H3S10 was investigated. In the context of SWI/SNF function it would be more logical to study regions of open chromatin, or nucleosome positioning. There are also several other histone modifications known to be associated with IEG induction so why not look at those?*

Response to point 3: Following suggestion of Reviewer #1 we have backed up our quantification of chromatin antibody labelling with a genome-wide study of accessible chromatin/DNA by performing an ATAC-Seq experiment. Our data clearly demonstrate an overall decrease of chromatin accessibility in *Brg1*^{D1Cre} mice in line with the increase of heterochromatin markers found in this model (DAPI, H3K9me3 and HP1 mentioned in point 5). This approach is detailed further in the answer to Point 6. Corresponding data are incorporated within the new version of our manuscript (Fig 7, Fig S10, S11 and S12).

Reviewer #1 wonders why we studied H3S10P. This histone modification is known to be involved in two structurally opposed processes: transcriptional activation and chromatin relaxation, and chromosome compaction during mitosis (Sawicka and Seier, *Biochimie* 2012). We chose to study this mark as it is induced in the NAc by cocaine treatment (Brami-Cherrier et al. 2005, cited in the text), and as, in fibroblasts, it leads to the promoter remodeling of some IEGs and onset of their transcription (Drobic et al. *Nucleic Acid Res.* 2010). Our interest was that this process involves MSK1 complexed with BRG1, and precedes SWI/SNF nucleosome remodeling, giving to transcription factors access to regulatory DNA sequences. We now mention this point in the result section, page 24.

Point 4. *“Evidence for antibody specificity should be provided for all antibodies. Ideally they should show lack of signal from a mutant tissue or cell line. An alternative would be to use blocking peptides. If this information is already available in the literature it should be cited.”*

Response to point 4: As suggested, we have cited the literature concerning the specificity of the antibodies we used in the material and method section and presented this information in a table (S1).

Point 5. *” The increase in intense regions of DAPI staining (representing heterochromatin) is interesting but does not confirm increased heterochromatin. How was this quantification done? It would need to be done on 3D images of sufficient resolution in the z plane to get an accurate measure of heterochromatin volume. There is not a clear explanation as to how the heterochromatin surface was measured or how the DAPI threshold intensity set to make those measurements? Were experimenters blinded when making these measurements. Alternate methods should be used to show expanding heterochromatin. Preferably ChIP-seq using known markers of heterochromatin. Alternately, immunostainings using known markers for heterochromatin.”*

Response to point 5: We now present the results of dense DAPI staining volume from 3D images instead of surface. This analysis has been performed from new sets of animals. Overall, we kept observing an increase of dense DAPI staining in *Brg1*^{D1Cre} mice compared to controls in all the tested conditions. Additionally, we performed H3K9me3 immunostainings, an epigenetic mark associated with heterochromatin and found results similar to that for the DAPI analyses (*i.e.*, increased levels in *Brg1*^{D1Cre} mice). These data are shown on Figure 6b and 6c. For these two staining, confocal images were acquired at high magnification (64x, zoom 4x), with a 1024x1024 pixel resolution (pixel size 0.06 µm) and a Z step of 0.25 idem. We quantified using open access Fiji plugin. The threshold was set visually for each nucleus so that all the “objects” were properly selected and differentiated. We ensured that the average threshold was the same for control and *Brg1*^{D1Cre} groups, as well as for treatment (undisturbed, acute, and repeated defeat). All the quantifications were made in a blinded manner. These details have been added in the material and method section on page 13.

We also quantified Heterochromatin Protein 1 (HP1) immunostainings, which are central units of heterochromatin packaging. The method of detection and quantification was different than for DAPI and H3K9me3. Indeed, HP1 immunostaining resulted in foci, as for BRG1 or BRM immunostainings. Hence, we quantified the number of foci using the same set threshold for all the nuclei. Here again, we found overall more foci in *Brg1*^{D1Cre} mice compared to control, especially after an acute defeat. These results are now included in Figure 6d.

Point 6. *"In general the connection between SWI/SNF, GR, and IEG expression is weak. This could be strengthened by more molecular data (ChIP) examining the interaction of these factors at spec"*

Response to point 6: We agree with reviewer 1 that the connection between SWI/SNF, GR and IEG expression is still not understood. In this paper, we addressed IEG expression first as a marker of neuronal activation to see whether, in absence of BRG1, as in absence of GR, MSNs neuronal activation was lowered. There is no clear evidence that GR through DNA-binding would be a direct regulator of *c-fos* or other IEG expression. We previously showed that in the hippocampus GR activates *egr1* expression through the activation of the MAPK pathway (Revest et al. 2005, *Nat. Neurosci.*). Concerning BRG1, previous work showed that it represses *c-fos* gene expression in adenocarcinoma cells (Murphy et al. *Mol Cell Biol*, 1999) and that the inactivation of BAF53 relieves IEG expression, including *c-fos*, in cortical cell cultures (Wenderski et al. 2020, *PNAS*). In MSNs, BRG1 and GR could act indirectly on IEGs expression, for instance by lowering the threshold of MSNs activation. Of note, we did not observe differentially accessible DNA intervals with low p-value within the *c-fos* gene in our ATAC-Seq data (see data file Fig7_genes) but observed one in the promoter of the *egr1* gene (p-value<0.01) that was more accessible in absence of BRG1 (see file dataset 3 Fig7 genes). We felt that addressing the mechanistic by ChIP seq was uncertain. Furthermore, addressing this way differential levels of fixation or modifications on IEGs on unsorted tissue is challenging as IEGs are only induced in a restricted number of MSNs. We therefore favored an ATAC-seq approach that gives a vision on a genome-wide scale of the differential accessibility of chromatin in the NAc of controls and *Brg1*^{D1Cre} mice. For some the genes whose chromatin accessibility is affected by the absence of BRG1, ChIP approach will be essential for our future work.

Response to reviewer #2 (Remarks to the Author):

"The manuscript is a sound study, based on substantial amount of interesting and novel data to reach the conclusion, i.e. the chromatin remodeller SWI-SNF are important epigenetic player in chronic social defeat (CSD) stress induced maladaptive changes in reward circuitry in depression.

The authors demonstrate how stress exposure induces stable reorganization of the transcription factor Glucocorticoid Receptor (GR) and chromatin remodeler subunit BRG1, in the nuclei of striatal medium spiny neurons (MSNs). They show that SWI-SNF complex subunits BRG1 and BRM are essential in dopaminergic neurons for the development of depression-like phenotype i.e. social avoidance and anxiety following repeated CSD. Lack of BRG1 and BRM in mutant mice was the reason for the attenuated responses to cocaine, as shown in locomotor sensitization and conditioned place preference (CPP) paradigms.

Behavioural changes or phenotype in animals that are exposed to chronic stressors and/or psychostimulants such as cocaine have been shown to be owing to the change in dopamine neurons. However, the absence of BRG1 or BRM in these models does not alter this neuronal population. They suggest that important action of BRG1 was through transcriptional response in striatal MSNs; they found that the induction of immediate early genes following Serine 10

phosphorylation on histone H3 on their promoters were significantly down on stress exposure resulting in heterochromatinization, thus affecting transcription.”

We thank Reviewer #2 for his comments and concerns. As described below in our point-by-point answer, we now provide additional experiments aimed at studying the relative contributions of BRG1 and BRM in all behaviors. We also demonstrated that Brg1 gene inactivation in the NAc is sufficient to reproduce *Brg1^{D1Cre}* phenotype following repeated social defeat. In addition, we provide a genome-wide analysis of chromatin/DNA accessibility in *Brg1* mutants and their controls using the ATAC-seq technology.

Response to specific points:

Point 1. *“The authors have shown that Brg1D1Cre mice do not show any abnormalities in learning and memory. However, they have not talked about the constitutive Brm knockout mice in this context. It would be interesting to know if any defects of learning and memory exist in these mice.”*

Response to point 1: As suggested by reviewer #2, we performed learning and memory tasks on *Brm^{-/-}* as well as on the compound mutant *Brg1^{D1Cre}:Brm^{-/-}* including working memory test and fear conditioning. We also performed tests of sociability following repeated social defeat in *Brg1^{D1Cre}:Brm^{-/-}* compound mutant mice. The data for the *Brm^{-/-}* mice have been included on figure S4a and c. We found that working memory relying on PFC and striatal function is only affected in compound mutants but not in *Brg1^{D1Cre}* and *Brm^{-/-}* mice suggesting that remaining SWI/SNF activity in these regions may be sufficient to ensure normal function. Fear memory, either contextual or cued, was not affected in any model. Looking closely at the data it seems that constitutive *Brm* inactivation may have a slight effect on contextual fear memory though it was still non-significant when pooling all the animals bearing this mutation (*i.e.* *Brm^{-/-}* and *Brg1^{D1Cre}: Brm^{-/-}* mice). These data can be explained by the fact that contextual and cued fear memory heavily rely on hippocampus and amygdala, two structures in which BRG1 remains expressed in our models. These results are now discussed on page 29.

Point 2. *“Under the results section titled ‘Brg1 gene ablation in dopamine-innervated areas prevents behavioural adaptation induced by repeated social defeat’, the authors should highlight the differences in anxiety-like behaviour observed between the Brg and Brm knockout mice. The authors might also consider revising the statement ‘Indeed, while repeated defeats significantly increased anxiety levels, no difference was observed between Brm^{-/-} mice and their control littermates’, since it does convey the result very clearly. Also, a possible explanation for this should be discussed. “*

Response to point 2: We now discuss further the differences observed between *Brg1^{D1Cre}* and *Brm^{-/-}* mice in anxiety-like behaviors and changed the sentence pointed by the reviewer #2 by the following one: *“Concerning the anxiety-like following repeated social defeat, mice deprived of BRM displayed a different phenotype than that of Brg1^{D1Cre} mice. Brm^{-/-} mice exhibited an increase of anxiety after repeated social defeats comparable to that of control mice whereas Brg1^{D1Cre} individuals did not (Figure 2h).”* page 19. We hope these modifications convey more clearly our result.

Point 3. *“The conditional place preference test was performed with the Brg1D1Cre mice, but not with Brm KO mice. The authors should either do the test or at least discuss its possible outcome, considering that Brg1 and Brm might have complementary roles.”*

Response to point 3: We had difficulties in generating animals in the past period, due to a strong breakdown of our animal facility in 2020. We chose to reserve the animals we obtained for the other experiments presented here. In our experience targeting MSNs, impaired locomotor sensitization to cocaine is generally associated with impaired place preference. We

thus assume that *Brm*^{-/-} mice would display no differences or a mild reduction of CPP. As suggested, this point is now mentioned in the result section page 21. The possibility of complementary functions for *Brm* and *Brg1* in behavioural responses to cocaine is now mentioned in the discussion section (page 28), together with the compensation between these two genes, observed in our rotarod and working memory experiments in which BRG1 or BRM activity in single mutants is sufficient for normal function whereas the simultaneous absence of both in compound mutants have a marked behavioral phenotype.

Point 4. *“The discussion should include a mention about the inhibitors of Brm and Brg1. Since a knockout of these genes confers stress resilience and reduced cocaine sensitisation, the possibility of using their inhibitors as therapeutic agents for mood disorders and addiction can be discussed.”*

Response to point 4: We mention the possibility of using inhibitors of BRM and BRG1 at the end of the discussion section on page 32. This point is especially interesting in the light of the additional results we now provide showing that adult inactivation of *Brg1* in the NAc of adult animals using AAV-mediated Cre expression also confers resilience to repeated defeats, excluding potential developmental effects of *Brg1* on the adult phenotype. These new data are included on Figure 2e and f.

Point 5. *“Overall, the manuscript is well written, but has some typos and grammatical errors which require proofreading.”*

Response to point 5: We proofread the manuscript and corrected typos and errors.

Response to reviewer #3 (Remarks to the Author):

“In this study, the authors examined the function of BRG1 and BRM, two ATPase subunits of the SWI/SNF chromatin complex, in stress behaviors within the reward pathway. Though chromatin remodeling was reportedly implicated in stress and addiction in the past, the role of BRG1 and BRM in this process is still elusive. The authors have carried out extensive behavioral, morphological, electrophysiological approaches, in addition to some other experimental assays, to illustrate the functional role of BRM and BRG1 in stress response. The data are interesting and the manuscript has an in-depth discussion. One limiting factor of this study is there is no direct examination of chromatin modification change at stress relevant genes. “

We thank Reviewer #3 for his comments. As requested by reviewers 1 and 2, we completed behavioral analyses in *Brm*^{-/-} and compound models. In addition, we demonstrated that *Brg1* gene inactivation in the NAc is sufficient to reproduce *Brg1*^{D1Cre} phenotype following repeated social defeat. We have now performed a direct examination of chromatin modification changes on a whole-genome scale in the NAc of controls and *Brg1*^{D1Cre} mice by using the ATAC-Seq technology. The data are presented in Fig 7 and Fig S10-S12. They reveal an overall decrease of chromatin accessibility in *Brg1*^{D1Cre} mice in line with the increase of heterochromatin markers found in this model. Among the DNA intervals that display differential accessibility to DNA, one mapped within a *Per1* promoter, a known GR target gene (p <5.0E-04) and two further intervals of lower statistical confidence matched with a region known to bind GR, as shown in a previous ChIPseq study from cultured mouse embryonic fibroblasts (see Figure 7). Other known GR target genes also display intervals with differential DNA access including *Fkbp5*, *Sgk1* and *Gilz* (*tsc22d3*).

Response to specific points :

Point 1. "The study is mainly based upon a conditional BRG1 knockout and a conventional BRM mutant mouse line. Do the two molecules have compensatory expression change when the other's transcription is decreased? I also feel the study on the BRM mutant mice does not add in much more insight. What are the differences between the two molecules? Do they function differently?"

Response to point 1: SWI/SNF complexes include either BRM or BRG1 as a catalytic subunit. Seminal paper showed that BRM knockout exhibit an increase of BRG1 protein levels in the brain. We also found a similar result with an increase of BRG1 in the NAc of *Brm*^{-/-} mice and identified an increase of BRM in the NAc of *Brg1*^{D1Cre} mice suggesting some degree of compensation at the protein level (see the graph below).

To explore potential compensation, we performed all the behavioral tests in our three models (*i.e.* *Brg1*^{D1Cre}, *Brm*^{-/-}, and *Brg1*^{D1Cre}: *Brm*^{-/-} compound mutants). The data are presented in Fig. S3 and S4 and suggest full functional compensation between BRG1 and BRM for locomotion (Fig. S3), motor learning (Fig. S4), and working memory (Fig. S4) as individual mutants have a behavior similar to that of control whereas compound mutant have a strong impairment. At the physiological level, we made a similar observation for VTA dopamine neurons activity. In contrast, for social avoidance following repeated defeats, each single mutant, and the compound one display similar phenotypes. Suggesting that each ATPase subunit is required for the induction of social avoidance. Finally, for locomotor sensitization to cocaine, the *Brg1* mutant displays a markedly impaired response, similar to that of the compound mutant whereas the *Brm* mutant displays no or mild impairment. Several possibilities could explain this result, including the possibility of complementary functions between *Brg1* and *Brm* both genes. These results are discussed page 28.

Point 2. "It seems BRG1 was solely knocked out in D1-MSNs, but not D2-cells, which have different roles in stress and addiction. The author should therefore specify their findings are to D1-MSNs only throughout the manuscript. Does this cre-loxp KO also exist in other brain regions? The authors should carry out a rescue assay to make the findings more conclusive."

Response to point 2: As now mentioned in the text, the TgYAC-D1aCre mouse line expresses the Cre in a population of MSNs that is larger than the canonical D1-expressing one. Indeed, in a previous study, we showed that in this mouse line, the Cre is expressed in around 90% of MSNs (Barik et al. 2013, cited in the text). Hence, this model cannot allow us to distinguish between the role of SWI/SNF in D1- vs D2-expressing MSNs. This is now more explicitly stated in our results section page 18 "with a deletion of *Brg1* in more than 90% of MSNs in the dorsal striatum and the NAc". It is also mentioned in the discussion on page 31 "It is of note that here we do not distinguish between the D1- and D2-receptor expressing MSNs and, although beyond the scope of our study, manipulating BRG1 independently in one

or the other neuronal population could yield distinct results as it has been shown for other factors. “

In addition to the striatum and the NAc, Cre expression is also found in deep cortical layers of TgYAC-D1aCre mice (Lemberger et al. 2007, cited in the text). To refine our findings, we used viral vectors to inactivate the *Brg1* gene only within the NAc of adult mice. This approach addresses two questions: is BRG1 in the NAc responsible for the phenotype observed in *Brg1*^{D1Cre} mice? Is the effect developmental or can it be induced by a loss of function at adulthood? The results show that the inactivation of the *Brg1* gene, restricted to the NAc at adulthood, also confers resilience to repeated social defeats. A similar effect of adult *Brg1* knock down has been observed in the context of cocaine responses (Wang et al. 2016, cited in the text). These results are shown in Figure 2e and f.

Point 3. *“It also appears the mice in this study were not congenic. Depends on the strain, the mice may have a mixture of J129, C57bl6, and or CD1. Have the authors tried backcrossing? The authors should examine and report the exact genetic background of each transgenic mouse line they used.”*

Response to point 3: *Brg1* and *Brm* mutant mice were bred on a mixed genetic background of two strains, 129SvEv and C57BL/6, with a higher contribution of C57BL/6, equivalent to a three-time backcross. In our experience, a major problem of mixed genetic background is the absence of response to treatment of control animals, as for locomotor sensitization to drugs of abuse. Our experimental animals clearly developed such a response. We however paid much attention to include as control animals the corresponding littermates of all mutant models. For locomotor sensitization to cocaine and motor learning experimental individuals were issued from first cousins from *Brm*^{wt/-}:*Brg1*^{+L2}:TgYACD1aCre x *Brm*^{wt/-}:*Brg1*^{+L2} crosses, only one control group was used. We now attempted to clarify this point in the material and methods section (page 5). The outbred CD1 genetic background is not present in our experimental animals. CD1 mice, that are more robust, were only used as “aggressors” in the repeated defeat paradigm to ensure that the issue of the encounter will be a defeat for our challenged individuals, and as interacting mice in social interaction tests. C57Bl/6J target mice were used for the three-chamber test.

Reviewers' comments:

Reviewer #2 (Remarks to the Author):

The revised version is satisfactory. The authors have answered most of my queries. from my side, it can be accepted for publication.

Reviewer #3 (Remarks to the Author):

The authors have addressed my prior comments. I do not have further questions.

REVIEWERS' COMMENTS

Reviewer #2 (Remarks to the Author):

The revised version is satisfactory. The authors have answered most of my queries. from my side, it can be accepted for publication.

Reviewer #3 (Remarks to the Author):

The authors have addressed my prior comments. I do not have further questions.

There is no further reviewers' comments